# Subjective optimality in finite sequential decision-making

**Yeonju Sin**[1☯], **HeeYoung Seon**[1☯], **Yun Kyoung Shin**[2], **Oh-Sang Kwon**[1]*,
**Dongil Chung**[1]*

**1** Department of Biomedical Engineering, UNIST, Ulsan, South Korea, **2** Department of General Education, University of Ulsan, Ulsan, South Korea

☯ These authors contributed equally to this work.
* oskwon@unist.ac.kr (O-SK); dchung@unist.ac.kr (DC)

**Data Availability Statement:** Analytic scripts and data are available on GitHub (https://github.com/dongilchung/secretary-problem).

**Funding:** This work was supported in part by the National Research Foundation of Korea (NRF-

## Abstract

Many decisions in life are sequential and constrained by a time window. Although mathematically derived optimal solutions exist, it has been reported that humans often deviate from making optimal choices. Here, we used a secretary problem, a classic example of finite sequential decision-making, and investigated the mechanisms underlying individuals' suboptimal choices. Across three independent experiments, we found that a dynamic programming model comprising subjective value function explains individuals' deviations from optimality and predicts the choice behaviors under fewer and more opportunities. We further identified that pupil dilation reflected the levels of decision difficulty and subsequent choices to accept or reject the stimulus at each opportunity. The value sensitivity, a model-based estimate that characterizes each individual's subjective valuation, correlated with the extent to which individuals' physiological responses tracked stimuli information. Our results provide model-based and physiological evidence for subjective valuation in finite sequential decision-making, rediscovering human suboptimality in subjectively optimal decision-making processes.

## Author summary

In many real-life decisions, such as hiring an employee, the current candidate is the only option decision-makers can choose among sequentially revealed options, while past options are forgone and future options are unknown. To make the best choice in such problems, decision-makers should set appropriate criteria considering the distribution of values and remaining chances. Here, we provide behavioral and physiological evidence for subjective valuation that explains how individuals set criteria deviating from optimality. The extent to which individuals expect from candidates, how sensitive they are to the value of candidates, and how costly it is to examine each candidate determine the way how individuals make choices. Our results suggest that seemingly suboptimal decision strategies in finite sequential decisions may be optimal in subjective valuation.

2018R1A2B6008959 to O.-S.K. and NRF-2018R1D1A1B07043582 to D.C.). The funders had no role in study design, data collection and analysis, decision to publish, or preparation of the manuscript.

**Competing interests:** The authors have declared that no competing interests exist.

## Introduction

Hiring a new employee is one of the toughest decisions to make as a team leader. Most of the time, there are only a limited number of job openings available and a limited time period in which to complete the hiring process. This process is even more difficult when applicants are accepted on a rolling basis, because one has to make a choice whether to accept the current applicant without knowing whether other future potential applicants would have been a better fit for the job. Likewise, there are many decision problems in life that are sequential and constrained by a certain time window. The 'secretary problem' is a classic example of this finite sequential decision problem and has been widely used to understand the optimal policy in making choices (e.g., to hire or not) under a limited number of opportunities [1,2]. Provided with the full information (i.e., distribution of candidates), the optimal solution for the problem is to choose the first number that is above a mathematically calculated decision threshold [3]. However, it is not clear whether and how humans deviate from optimal choices. Here, we used one variant of the secretary problem [4], in which the distribution of candidates is given and the reward is the value of the chosen candidate, to investigate (i) whether individuals make the optimal decision in a finite sequential decision problem, and (ii) if not, how do they make their decisions. Our results provide behavioral and physiological evidence supporting that individuals make threshold-based choices in a finite sequential decision problem and that seemingly suboptimal decision patterns (deviation from the optimal) originate from the process of optimally calculating thresholds using individuals' subjective value function.

Since the 1960's when the secretary problem was originally introduced, a vast amount of studies were conducted examining the optimal solution of the problem [2–5]. Although an equally large volume of studies followed focusing on biases in human choice patterns, the major stream took a descriptive approach explaining the extent to which human choices drifted apart from the optimal solution [6–9], leaving where the bias originated from an open question.

In various psychological studies, including an optimal stopping problem in the mate choice domain [10] and consumers' purchasing decisions [11,12], it was suggested that individuals' heuristic valuation, the framework of Prospect theory [13], underlies the biases observed in their decision patterns [14]. Specifically, the Prospect theory has shown that individuals' choices are guided by subjective valuation of potential gains and losses relative to the context where they are situated [13]. We constructed a computational model that adopts this concept of 'reference point' and examined whether the process of subjective valuation explains how individuals' decision processes deviate from the optimal decision strategy. In addition, to capture individual differences in subjective valuation, we hypothesized that nonlinear value sensitivity (i.e., concave value function for gains and convex value function for losses) [13] would take crucial parts in determining individuals' decision patterns.

To examine how individuals make choices in a finite sequential decision problem, we recorded behavioral choices and response time (RT) of 87 participants (male/female = 43/44, age = 22.74 ± 1.98 years) as they made a series of choices to accept or reject a random number presented on the screen (Fig 1A). During each round, they had a fixed amount of opportunities (chances) to evaluate a new random number by rejecting previously presented numbers. When they accepted, the presented number was added to their final payoff, and then they moved on to the next round (up to 200 rounds) that consisted of a new set of chances. We implemented two separate experiments. In Experiment 1, participants had up to five opportunities (K = 5), and they were not explicitly informed of the maximum number that would be presented. Experiment 2 tested three distinctive contexts where participants had up to two,

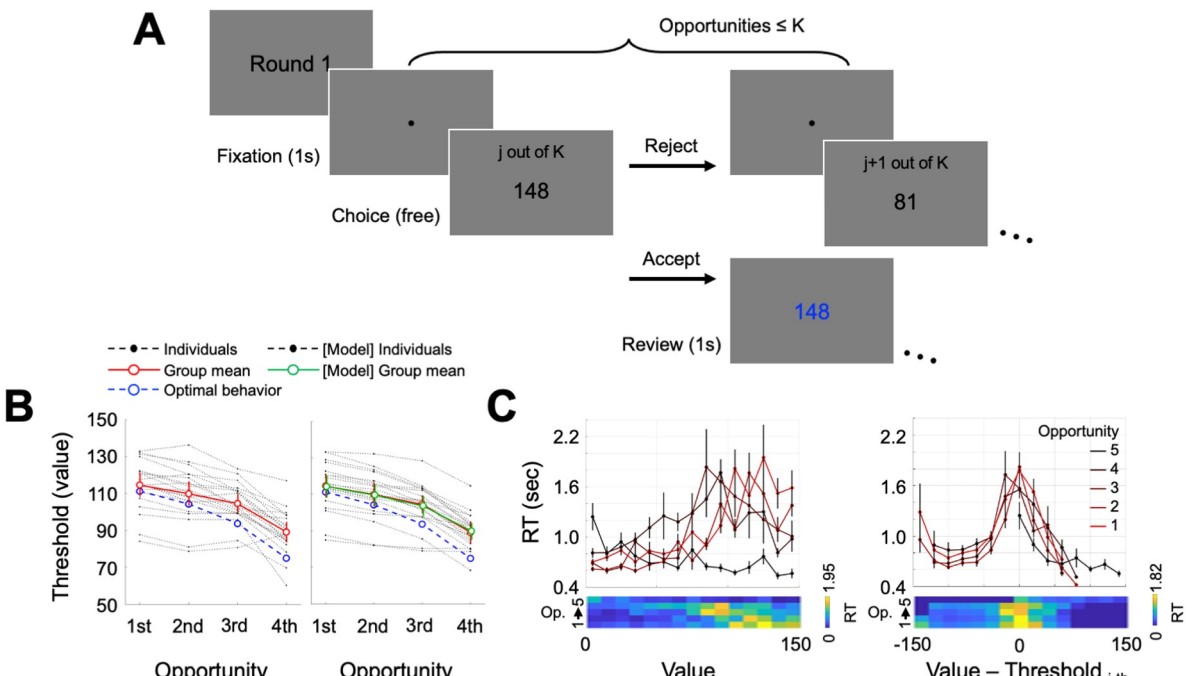

**Fig 1. Experimental procedures and behavioral results of Experiment 1. (A)** Participants made a series of choices between accepting and rejecting a presented number. At each round, they had up to K opportunities (K = 5 in Experiment 1, K = 2 or 5 in Experiment 2) to reject the number and get a new random number; the round ended when participants accepted a presented number. At the last opportunity, participants were given no choice but to accept the presented number. A new set of stimuli (numbers) was used in the next round. **(B)** The optimal decision threshold per opportunity (blue), calculated under the assumption of the full information, was compared with a corresponding empirical decision threshold (red). **(C)** Response times (RTs) for each opportunity were computed against the presented stimuli values. Regardless of the opportunity, RTs showed negative association with the absolute distance between the presented stimuli and the corresponding decision threshold. That is, participants showed the shortest RTs for the numbers that are farthest from decision thresholds, and vice versa. Error bars represent s.e.m.

five, or ten opportunities (K = 2, 5, or 10), and the participants were informed of the full distribution information (including the maximum).

Various theoretical and experimental studies took into account physical and/or mental efforts as a cost (or cost function) in subjective valuation [for review, see 15,16]. In line with the literature, we assumed that individuals would take into account a mental cost (referred to as 'waiting cost' hereafter) in valuation processes. Such an assumption is also linked with previous other studies where a version of secretary problem is used with their focuses being on the impact of sampling cost [8,17,18]. Thus, we also included a waiting cost [8,18] parameter as a linear disutility (negative subjective value), which captures a broad range of mental costs during decision processes (see Methods for detailed model structure; see **Text A in** S1 File for detailed model justification).

In conjunction with the computational modeling approach, we recorded individuals' pupil dilation responses and analyzed their association with participants' choice behaviors to corroborate our suggested model. A rich set of evidence suggests that pupil dilation (or contraction)—a neurophysiological signature coinciding with norepinephrine levels in the brain—reflects not only individuals' arousal level [19,20], but also cognitively complex information, such as value [21,22], uncertainty [19], surprise [23], cognitive conflict [24,25], and choice [26]. More specifically, recent studies showed that pupil-linked neuromodulatory systems have a critical role in decision formation, rather than simply in reflection of the decision

consequences [26], and moreover that the size of pupil responses is associated with individuals' choice patterns [27,28].

Based on these previous studies, we hypothesized that individuals' pupil responses would be linked to their decision processes in three folds; i) pupil responses may be larger to the options that will be accepted versus rejected, ii) pupil dilation may reflect the deviation between option values and decision thresholds, supporting value- and threshold- based decision processes, and iii) pupil responses in individuals who show larger value sensitivity may be larger. To test these hypotheses, we conducted an additional experiment (Experiment 3) where actions (choosing to accept or reject) were temporally dissociated from physiological responses to stimuli; participants (N = 24, male/female = 12/12, age = 22.67 ± 2.28 years) were not allowed to make a choice until an audio cue was played. All the other settings were equal to Experiment 2 where participants had up to five chances (K = 5). Non-overlapping samples were obtained from each experiment (see Methods for detailed procedures).

## Results

### Experiment 1

**Individuals show higher decision thresholds than the optimal decision model.** In Experiment 1, participants had up to five opportunities within each round to examine a presented number and make choices either to accept or reject. Each presented number, sampled from a uniform distribution ranging from 0 to 150, could be considered as an option whose value matches its face value (the number). Note that here in Experiment 1, participants were not explicitly informed of the distribution (including the maximum) from where the numbers were extracted. Because individuals can only accept a single number within each round, they should accept a number only when it is large enough. Specifically, an optimal decision-maker should not accept a presented number unless it is larger than the expected value of successive opportunities. For example, individuals should accept any numbers at the last opportunity (i.e., the fifth opportunity in Experiment 1) and thus the expected value of the last opportunity is 75. Based on this information, at the opportunity one before the last (the fourth in Experiment 1), a value-maximizing individual should accept any numbers higher than 75 but reject other numbers. Following the dynamic programming approach [29], we computed an optimal threshold for each opportunity (Fig 1B, blue).

To examine whether individuals follow such decision processes, we calculated empirical thresholds—the value where individuals were equally likely to accept or reject—at individual level using each individuals' behavioral choices (male/female = 10/10, age = 22.85 ± 1.31 years) (Fig 1B). Consistent with the optimal thresholds (blue), empirical thresholds (red) at the later opportunities were lower than those at the earlier opportunities (mean threshold differences between the first and the second = 4.70, t(19) = 5.24, Cohen's d = 1.17, p = 4.66e-5; the second and the third = 5.39, t(19) = 3.99, Cohen's d = 0.89, p = 7.84e-4; and the third and the fourth = 15.38, t(19) = 7.10, Cohen's d = 1.59, p = 9.41e-7). However, participants showed empirical thresholds significantly higher than the optimal thresholds, indicating that people have higher expectations about later opportunities (the difference between empirical and optimal thresholds = 8.49, t(19) = 3.28, Cohen's d = 0.73, p = 0.004).

Compared with optimal thresholds, it is not difficult to notice that the average empirical thresholds have a shallower slope as evidenced by the increasing difference between the empirical and optimal thresholds across opportunities (mean slope of [empirical—optimal]: 3.75, t(19) = 4.40, Cohen's d = 0.98, p = 3.08e-4). Although the empirical decision thresholds suggest otherwise, one may still suspect that an alternative heuristic individuals might have used was to apply a constant threshold regardless of the number of remaining opportunities (i.e.,

applying a constant threshold across all opportunities). It is well known that easier choices—here, deciding whether to accept or not the presented value that is far smaller or larger than the threshold—require shorter response times (RT) [30,31, see 32 for review]. If individuals applied the same threshold across all opportunities, mean RTs should be symmetric around a fixed value (i.e., fixed threshold), whereas if individuals applied differential thresholds at each opportunity, RTs should be symmetric around the thresholds that correspond to the opportunity. To examine this possibility, we calculated mean RT within each opportunity. The symmetric pattern was observed only when mean RTs were calculated as a function of presented values adjusting for the estimated empirical threshold within each corresponding opportunity (Fig 1C). This result suggests that individuals did apply differential thresholds for each opportunity during decision-making.

**Subjective optimality explains individual choice patterns.** Prospect theory has suggested that outcomes are perceived as gains and losses relative to a certain reference point, and that gains and losses are valued following concave and convex subjective value functions, respectively [13]. We drew on this framework to evaluate potential decision processes accounting for individuals' sub-optimal decision thresholds. In accordance with Prospect theory, we hypothesized that individuals' subjective valuation (Util) for a given value (v) is dependent on their individual reference point (r) and nonlinear value sensitivity (ρ), as follows:

$$\text{Util} = (v - r)^\rho \text{ if } v \geq r$$

$$\text{Util} = -(r - v)^\rho \text{ otherwise.}$$

In the current study, we assume that individuals would set an expectation about future outcomes and use the expectation as a reference point [33] (see **Text A in** S1 File for detailed model justification). Note, we focused on valuation per se, and thus, the time it took for individuals to establish (learn) their reference points (their own perspective of the environment) was assumed negligible (**Figs H and I in** S1 File; see Discussion for further consideration of learning effects). Importantly, two additional components were introduced. First, individuals may perceive the waiting time until acceptance costly and take it into account in valuation [8,18]. Second, we hypothesized that this subjective value-based computation occurs not only during active decision-making, but also at mental simulation such that individuals use their subjective valuation in constructing expectations of each opportunity (i.e., computing decision thresholds; Fig 2A).

This 'Subjective optimality model' with a waiting cost converges to three nested models in special cases: the Subjective optimality model without a waiting cost (Cost = 0), the Optimal decision model (ρ = 1), and the Constant threshold model (ρ = 0) (see Methods for model details). A formal model comparison using Akaike Information Criterion (AIC) revealed that the Subjective optimality models with and without a potential waiting cost showed superior explanatory power compared to the other two nested decision models (**Fig A in** S1 File). Between the two Subjective optimality models, the model without a waiting cost showed a better model-fit than the model with a waiting cost (95% CI of ΔAIC = [-2.082, -0.099]; see **Fig A in** S1 File). These results suggest that the waiting cost was negligible in Experiment 1 and that individuals used the reference point significantly larger than zero (mean r = 110.51, **Fig G in** S1 File), which set values larger than the reference point as gains, and any value stimuli smaller than the reference point as potential losses (Fig 2B). Moreover, this result indicates that individuals use marginally diminishing (concave) and increasing (convex) subjective value function for gains and losses, respectively, in finite sequential decision-making.

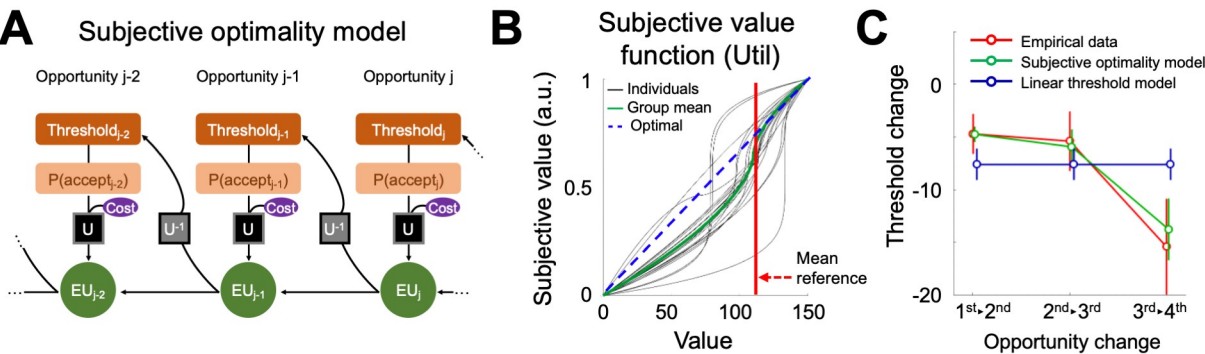

**Fig 2. Subjective optimality model. (A)** The optimal decision model assumes that individuals compute the decision threshold of a certain opportunity based on the expected value of successive opportunities. In the 'Subjective optimality model', expected values of the successive opportunities are replaced by expected utilities (EU) calculated based on the subjective value function as per Prospective theory. **(B)** Two free parameters, reference point, and nonlinear value sensitivity define subjective valuation of the presented stimuli values. Group average subjective value function (green) is depicted using the group mean of individual estimates: reference point = 110.51; value sensitivity = 0.62. **(C)** Decision thresholds calculated from empirical data showed a decreasing pattern along the opportunities, but moreover, the extent of threshold change between opportunities also showed a decreasing pattern (i.e., second derivative of the decision threshold curve < 0). Such a change in the slope of decision threshold curve is a unique feature that our suggested Subjective optimality model successfully could explain (differences between the model and empirical data do not vary across opportunities; $F(2, 38) = 0.70$, $\eta^2 = 0.036$, $p = 0.090$), and has superior explanatory power to the linear threshold model. Negative values on the y-axis indicate that decision thresholds decrease between opportunities. Error bars indicate s.e.m.

From the formal model comparison, we noticed that our suggested computational model showed comparable model-fit with some of the alternative models including the Linear threshold model, a model recently presented to well explain human stopping decisions in sequential decision-making [34]. Here, besides examining model fits, we examined whether the distinctive linear feature of the Linear threshold model presents in our empirical data. Decision thresholds calculated from empirical data showed a decreasing pattern along the opportunities (Fig 1B), but moreover, the extent of threshold change between opportunities also showed a decreasing pattern (i.e., second derivative of the decision threshold curve < 0; Fig 2C). Such a qualitative difference between the linear threshold model and the empirical data was corroborated by the observed statistical difference across opportunities ([1st-2nd] vs [2nd-3rd] vs [3rd-4th]; $F(2, 38) = 14.13$, $\eta^2 = 0.43$, $p = 2.59e\text{-}05$). These results suggest that individuals' choice data in the current variant of Secretary problem task cannot be fully explained by the Linear threshold model.

## Experiment 2

**The Subjective optimality model predicts behavioral alterations in the context of scarce opportunity.** In our suggested model, a change of reference point reframes one's subjective valuation and, in turn, alters decision thresholds. As described earlier, we assumed that individuals would set an expectation about future outcomes based on the number of opportunities they have, rather than trial-by-trial experience (c.f., [35]), and use the expectation as a reference point [33]. That is, we predicted that, based on our suggested model, one would set a lower (or higher) reference point in the context where one expects less (more) overall outcomes, which consequently lower (heighten) their decision thresholds. To examine whether empirical data matches the prediction from the model, we conducted a second experiment using a between-group design where one group of participants had two (K = 2), the second group of participants had five (K = 5), and the third group of participants had ten opportunities (K = 10) in each round (Fig 1A). For all three separate task conditions (K = 2, 5, and 10),

participants were informed that the maximum possible number that would be presented was 150, and any numbers between 0 and 150 (inclusive) were equally likely to appear.

Individuals who had up to two opportunities (K = 2) always had to accept the second value if they rejected the first presented stimulus. In the Optimal decision model, decision thresholds are determined by the number of remaining opportunities and expected values calculated following the dynamic programming approach [29]. This means that regardless of the total number of opportunities (K = 2, 5, or 10), individuals should be facing the same problem at the opportunity preceding the last (i.e., the first opportunity in K = 2, the fourth in K = 5, and the ninth in K = 10) if they were following the Optimal decision model. On the contrary, the Subjective optimality model predicts differently. The decision thresholds depend on individuals' reference point at a given context, which is dependent on the total number of opportunities and the expected future earnings in the context.

As predicted from the Subjective optimality model, the decision thresholds estimated from participants (K = 2: N = 23, male/female = 11/12, age = 23.09 ± 2.09 years; K = 5: N = 21, male/female = 11/10, age = 23.19 ± 1.86 years; K = 10: N = 20, male/female = 10/10, age = 22.00 ± 2.41; non-overlapping from Experiment 1) were significantly different depending on the number of opportunities one had per round (threshold$_{K = 2, 1st}$ = 79.28 ± 15.45, threshold$_{K = 5, 4th}$ = 90.42 ± 9.72, threshold$_{K = 10, 9th}$ = 93.50 ± 17.77; F(2,61) = 5.69, $\eta^2$ = 0.157, p = 0.0054; post-hoc comparison K = 5 vs 2: CI = [0.49, 21.78], p = 0.038; K = 10 vs 2: CI = [3.44, 25.00], p = 0.0067; K = 10 vs 5: CI = [-7.93, 14.10], p = 0.78; **Fig H in** S1 File). Note that decision thresholds and response time patterns for Experiment 2 (K = 5) were comparable with those of Experiment 1 (Fig 3A and **Fig B in** S1 File), indicating that informing the full distribution information to participants did not have any noticeable effects. Furthermore, a model comparison using AIC scores revealed that the Subjective optimality models with and without a potential waiting cost explained individuals' choice behaviors comparably well (95% CI of ΔAIC = [-2.047, 3.519]), but significantly better than other models (see **Fig A in** S1 File for model comparison results).

Next, we examined whether our model quantitatively captures behavioral alterations dependent on the scarcity or abundancy of opportunities. Again, we hypothesized that the changes in the number of opportunities and the corresponding changes in expected future outcomes would manifest the adjustments of reference points. To compute the prediction, we simulated choice behaviors by adjusting the reference point parameter while keeping all the other parameters estimated from the empirical data in K = 5. As depicted in Fig 3B, the model predicts that the decision thresholds at the opportunity preceding the last will vary accordingly. Consistent with our prediction, the observed behavioral thresholds closely matched the model-based threshold predictions for K = 2 (78.50 ± 4.09) and K = 10 (92.55± 3.58) (see Methods for model prediction details). Note that a direct parameter estimation from the empirical data revealed that individuals' characteristics (e.g., value sensitivity) in subjective valuation other than the reference point were indeed comparable between the task conditions, consistent with the assumption we made for the model-based threshold prediction (Fig 3C **and Fig G in** S1 File). The performance of model-based prediction showed comparable results when the empirical data in K = 10 were used to predict decision thresholds for K = 2 and 5 (**Fig C in** S1 File). Besides the prediction approach, we also confirmed that empirical decision threshold curves from Experiment 2 also showed negative second derivative across contexts, and that Subjective optimality model captures the patterns (Fig 3D). These results suggest that change of the decision context indeed alters individuals' reference point and their choice patterns, and that the reference point has a critical role in finite sequential decision-making.

One may suggest that the task with K = 2 is simple enough for participants and that they would have followed the optimal strategy (i.e., using 75 as a decision threshold at the first

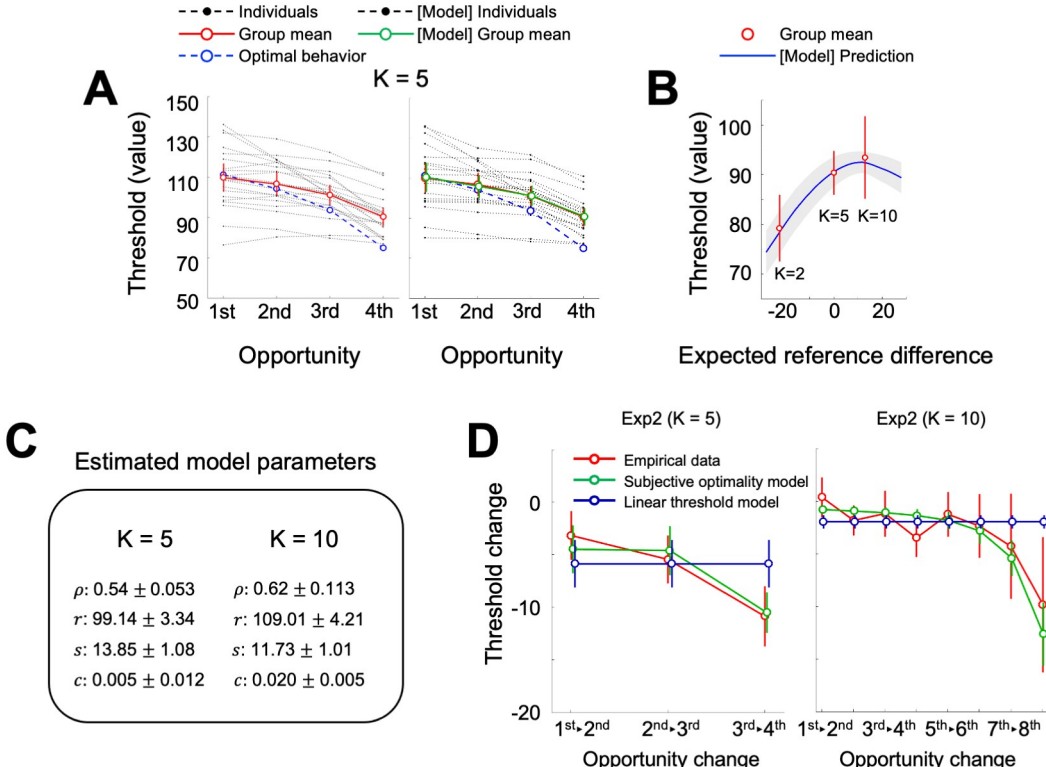

**Fig 3. Behavioral results of Experiment 2. (A)** In individuals who had five opportunities (K = 5), empirical decision thresholds (red) along the opportunities were comparable with that of Experiment 1. (B) To examine whether or not our Subjective optimality model can be generalized to other contexts, a model prediction of the decision threshold was made for K = 2 and 10; value sensitivity was assumed to be the same even in the different context, but the reference point was adjusted proportionately to the changes of the expected payoff. Empirical (observed) decision thresholds (red) at the opportunity preceding the last in the condition with two (K = 2), five (K = 5), or ten (K = 10) opportunities were consistent with the prediction (blue). Error bars represent 95% confidence interval. (C) Model parameters were estimated using the Subjective optimality model with a waiting cost. The parameters cannot be estimated for K = 2, because there is only one valid opportunity per round for each individual. **(D)** The pattern within the decision curve (**c**hange in the slope of decision threshold) was consistent in both of the Exp 2 data, and our Subjective optimality model successfully explained the pattern (differences between the model and empirical data do not vary across opportunities; Exp2 (K = 5): $F(2, 40) = 2.37$, $\eta^2 = 0.11$, $p = 0.11$; Exp2 (K = 10): $F(7, 119) = 0.67$, $\eta^2 = 0.039$, $p = 0.70$), and has superior explanatory power to the linear threshold model. Negative values on the y-axis indicate that decision thresholds decrease between opportunities. Error bars indicate s.e.m.

opportunity). However, this is unlikely given that only 7 out of 23 participants' credible intervals of the empirical decision threshold, defined by the 95% highest density interval, included 75 (see Methods). Furthermore, the large across-individual variability in behavioral decision thresholds ($SD_{K = 2} = 15.45$; Fig 3B) showcased that the Optimal decision model cannot explain individuals' decision strategies. These results again support the Subjective optimality model suggesting that individuals make threshold-based choices in a finite sequential decision problem, and that seemingly suboptimal decision patterns (e.g., waiting for future chances) may have originated from the process of calculating thresholds using individuals' subjective value function.

## Experiment 3

To further investigate physiological instantiation of the decision processes implemented in our model, we examined changes of pupil diameter acquired while participants made a series of

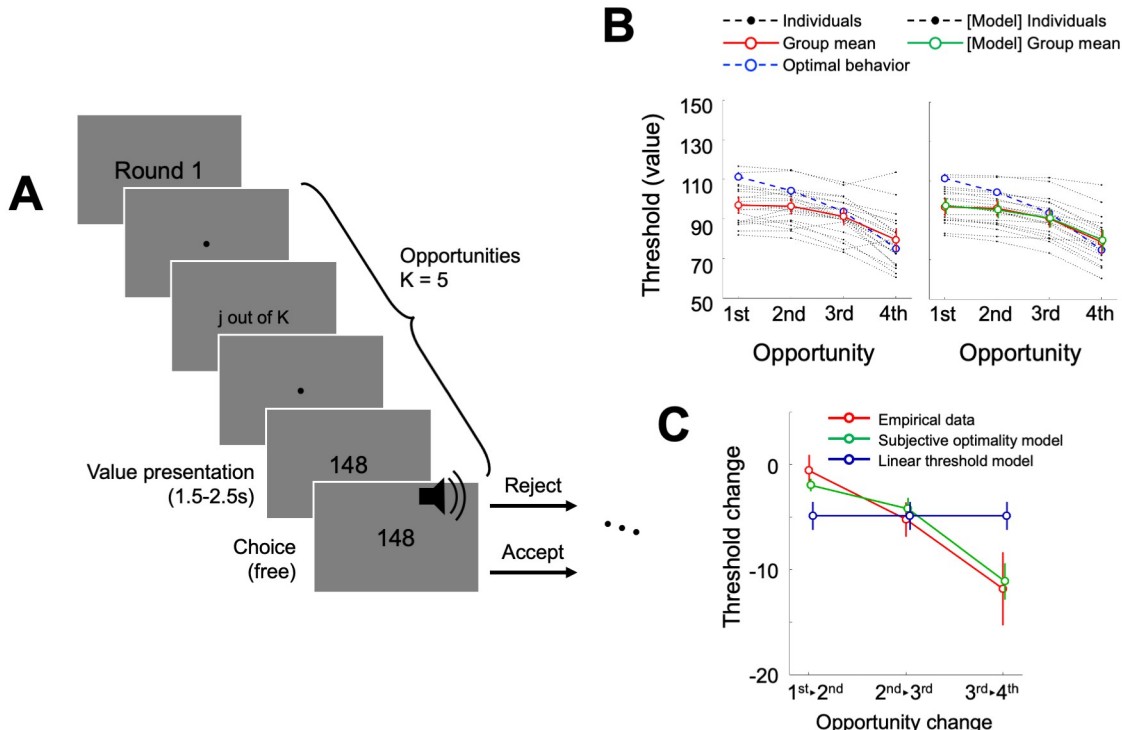

**Fig 4. Experimental procedures and behavioral results of Experiment 3. (A)** To temporally dissociate valuation from action selection, we implemented a modified task design where individuals had to wait for an audio cue to make choices. **(B)** Empirical decision thresholds (red) were compared with the optimal decision thresholds (blue). Compared with Experiments 1 and 2, in Experiment 3, individuals showed lower decision thresholds at the early opportunities. Error bars represent s.e.m. **(C)** The pattern within the decision curve (change in the slope of decision threshold) was consistent in Exp 3 data, and our Subjective optimality model successfully explained the pattern (differences between the model and empirical data do not vary across opportunities; $F_{(2, 40)} = 1.80$, $\eta^2 = 0.082$, $p = 0.18$), and has superior explanatory power to the linear threshold model. Negative values on the y-axis indicate that decision thresholds decrease between opportunities. Error bars indicate s.e.m.

choices. As noted earlier, we drew upon previous findings suggesting the role of pupil-linked neuromodulatory systems in decision formation [26] and the association between pupil responses with individuals' choice patterns [27,28], and hypothesized that participants' pupil responses would reflect their subjective valuation defined within our suggested computational model: i) deviation between option values and decision thresholds, ii) acceptance versus rejection decisions, and iii) individuals' value sensitivity. To test this hypothesis, we operated a slightly modified task; i) participants had to view the presented stimuli for a period of time (1.5–2.5 seconds) before being allowed to accept or reject the stimuli, and ii) an audio cue was used to announce to participants that they could make a choice (Fig 4A). This modification temporally dissociated choice from other cognitive processes (e.g., valuation) and prevented the introduction of any visual confounds in analyzing physiological signals at the time of decision-making. Participants had up to five opportunities per each round, and all other experimental settings were equal to Experiment 2; the full distribution information was informed to participants, and all participants played up to 200 rounds (see Methods for details).

**Waiting is costly.** Twenty-one new participants were recruited for Experiment 3 (10 females, age = 22.62 ± 2.38 years; non-overlapping from Experiments 1 or 2). With the addition of forced waiting time, which accumulated over opportunities, we expected that participants would perceive choices to accept after a longer wait less valuable [36,37] and thus, they would accept earlier. Consistent with our expectation, a stark difference in the behavioral

pattern was observed in Experiment 3 compared to Experiments 1 and 2. Specifically, decision thresholds from empirical data in Experiment 3 (red solid line) were below the optimal decision thresholds (blue dotted line), indicating that participants were more likely to accept small numbers that they would have rejected in the other two experimental settings (Fig 4B).

This result was corroborated by the model-based results. First, the Subjective optimality model with a waiting cost showed superior explanatory power for Experiment 3 compared with alternative models (**Fig A in** S1 File), emphasizing again that the waiting cost plays an important role in finite sequential decision-making [8,18]. In addition, as in both Experiments 1 and 2, empirical decision threshold curve showed negative second derivative, which is the feature that Linear threshold model, unlike our Subjective optimality model, cannot capture (Fig 4C). Second, the average of the estimated waiting cost parameter was significantly larger than zero in Experiment 3 ($t(20) = 63.51$, Cohen's $d = 13.86$, $p = 1.51e-24$), and it was larger than the cost parameters in the other experiments (Experiment 3 > 1: $t(40) = 15.67$, Cohen's $d = 4.84$, $p < 1.00e-15$; Experiment 3 > 2 ($K = 5$): $t(41) = 5.64$, Cohen's $d = 1.72$, $p = 1.41e-6$) (**Fig G in** S1 File). Third, as it was intended from the task modification, individuals' behavioral change was sourced specifically back to the waiting cost parameter, such that other parameters (nonlinear value sensitivity and reference point) were not affected (**Fig G in** S1 File). These results together support our interpretation suggesting that the perceived cost of waiting underlies the behavioral alteration in the new task environment.

**Pupil dilation reflects choice and decision difficulty.**   As described above, we then examined whether physiological responses reflect cognitive decision processes. First, we compared pupil diameter changes between accepted and rejected opportunities. Consistent with previous reports, pupil size was significantly different depending on the subsequent choices [26] (Fig 5A). Particularly, pupil dilations within 558–726 msec and 1182–1500 msec were associated with subsequent acceptance of the presented values ($t(17) > 2.11$, all ps < 0.05). Only the latter cluster remained significant after correcting for multiple comparisons using a cluster-based permutation method [38] ($p_{corrected} = 3.50e-4$). Still, given the fact that the time of the earlier cluster (558–726 msec) overlaps with the range of RTs in Experiments 1 and 2 (Fig 1C **and Fig A in** S1 File), this result suggests that participants may have covertly made choices as early as 550 msec and the cognitive process was reflected in the physiological responses [26] (see **Fig D in** S1 File for a pupil size result reflecting individuals' arousal level).

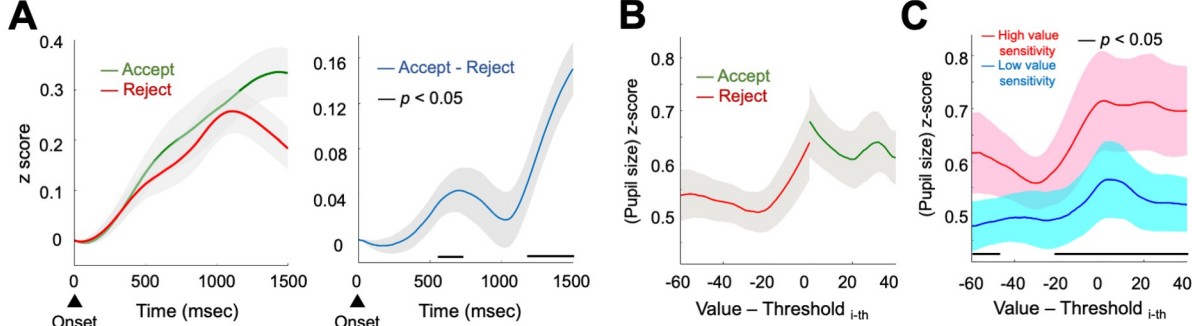

**Fig 5. Pupillometry responses reflect subsequent choices and decision values. (A)** Pupil size change from the stimuli onset was measured, separately for the accepted (green) and rejected (red) opportunities. Paired comparison between the cases revealed significant pupil dilation for the accepted stimuli at the early stage after the onset, and again at the later time. **(B)** To examine whether or not pupil size reflected stimuli value, the peak pupil size between the stimuli onset and 1500 msec after the onset was depicted as a function of the signed distance between stimuli value and the corresponding decision threshold. **(C)** Individuals who had higher value sensitivity in their estimated parameter (median split; red) showed more pronounced pupillometric responses reflecting the value information. Shades represent s.e.m.

Next, we calculated the peak pupil diameters between the stimuli onset and 1500msec after the onset as a function of stimuli values. This was done for accepted and rejected stimuli separately, so that the relationship between pupil sizes and values is independent of subsequent choices. As we observed from RT patterns (Fig 1C **and Fig A in** S1 File), pupil size was positively correlated with 'decision difficulty'. That is, pupil size decreased as a function of the absolute distance between the decision threshold and value of the presented stimuli (slope = -0.0105, t(17) = -3.15, Cohen's d = -0.74, p = 0.0059; Fig 5B). A comparable result was observed across all the opportunities (**Fig E in** S1 File) or when we used pupil diameter at 1500msec after the onset, the latest time point when no single participant was allowed to make choices by the task design. Together, these results suggest that pupil dilation reflects both decision difficulty and subsequent choices [24,26], the two crucial components comprising subjective valuation [39,40].

**Physiological sensitivity matches behavioral value sensitivity.**   As evidenced by the model parameter estimates, there are individual differences in the extent to which one responds to a unit increase of presented stimulus value (i.e., value sensitivity). We tested whether or not this modeling construct of individual characteristics matches with individuals' physiological responses. To provide an illustrative description, we divided participants into two subgroups based on their parameter estimation (median split) where one group had lower value sensitivity and the other group had higher value sensitivity. For each group, we calculated average pupil dilation as a function of signed decision difficulty (the difference between stimulus value and the decision threshold of the corresponding opportunity) (Fig 5C). Individuals who had high value sensitivity (red) showed relatively high pupil dilation compared to individuals who had low value sensitivity (blue). This positive correlation between value sensitivity and pupil dilation was statistically significant within the signed decision difficulty (Value–Threshold$_{i-th}$) ranging from -60 to -47 and from -22 to 40 (Pearson's correlation r > 0.47, all ps < 0.05). Only the latter cluster remained significant (p$_{corrected}$ = 0.016) after correcting for multiple comparisons using a cluster-based permutation method [38]. The result indicates that individuals who have high behavioral value sensitivity indeed have higher physiological sensitivity to stimuli value. Moreover, the consistent patterns across physiological and behavioral data reflecting individuals' characteristics serve as additional evidence suggesting the use of subjective valuation in finite sequential decision-making.

## Alternative mechanisms and factors

In the current study, we tested our hypothesis that individuals use subjective valuation and dynamic programming approach during finite sequential decision-making. Here, we further explored potential factors (e.g., regret, selection bias) that could be recruited and contribute to biasing behavioral choices.

**Regret model.**   It is possible that individuals may regret about the past choices, and thus less likely to accept the stimuli values they previously rejected. To more directly examine potential effects of "regret", we simulated behavioral choices with additional computational model assuming that participants would accept the candidates those are higher than the highest value they have rejected before (see Methods for simulation details). The Regret model partially explain the observed empirical data (Fig 6A). The accumulated impacts of regrets, as expected, is the part that allowed decision thresholds to have shallower slopes than the Optimal. Note that the current Regret model assumed perfect memory, which simulates maximum effect of previous stimuli value, and thus, was provided with the maximal power to explain "flat" decision thresholds across opportunities induced by the regret. Our simulation results suggest that, although we cannot rule out the impact of regret (see Discussion for further

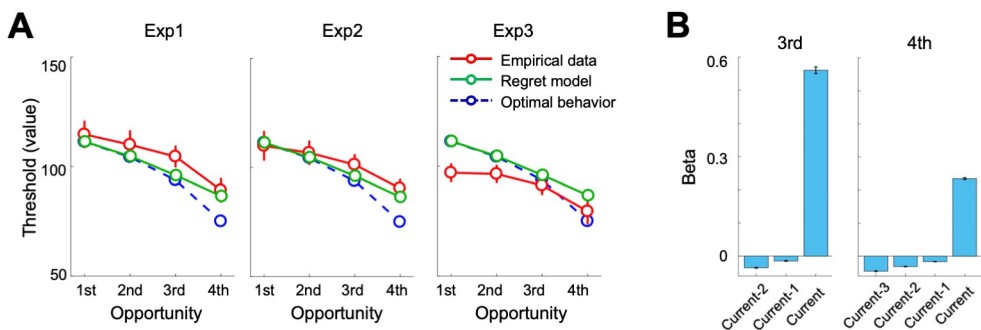

**Fig 6. Subjective optimality model versus an alternative Regret model.** As depicted in **Fig I in** S1 File, there are potential effects of past stimuli values on the subsequent choice. **(A)** To more directly examine potential effects of "regret", individuals accept less likely the stimuli values that are above the optimal threshold but below a value which they have previously rejected, decision thresholds simulated with a regret model are compared against the optimal decision model and empirical data. Error bars indicate s.e.m. **(B)** Effects of past stimuli values on the current decision were examined by applying the logistic regression analyses using the simulated data from the regret model. The regret model predicts meaningful deviation from the optimal model in the 3rd and 4th opportunities, and the deviation can be attributed to the negative weight on the preceding opportunities. However, the regret model, inconsistent with the observed pattern from the empirical data (**Fig I in** S1 File), predicts that the value in the earlier opportunities will affect the current decision more than the value in the recent past. Error bars indicate s.e.m.

discussion about potential learning effects), the regret is not the major factor that underlies the behavioral suboptimality.

**Selection biases.** It is possible that due to decision variabilities, participants' choice rounds where they happened to have lower decision thresholds would be accepted at earlier opportunities, and thus be under-represented at subsequent opportunities. We ran a new simulation to implement how the behavioral pattern would become if the selection bias indeed is the major factor that biases individuals away from the optimal behavioral pattern (see Methods for simulation details). To test whether selection bias explains the flatness of the empirical decision thresholds, we first, as the most conservative setting, assumed that the entire behavioral choice variability contributes to the within-subject variability across rounds (see Methods for justification). Based on the simulated choices, the model that explicitly incorporated selection bias ('Selection bias model' hereafter) partially explained the deviation from the optimal thresholds and show relatively shallower slopes as expected (Fig 7A). However, compared to empirical decision thresholds across opportunities, the simulation slopes were relatively steeper even though the simulation was run with the maximum level of variability as we described above. Note that in this simulation-based analysis, we are using the differences in slopes across decision thresholds, rather than other features (e.g., the deviation at the initial opportunities in Experiment 3), as criteria to examine whether the simulation model captures the core mechanism explaining the empirical data.

Although our simulation using the maximum level of within-subject variability across rounds was our attempt to give the Selection bias model the maximal power to explain "flat" decision thresholds across opportunities, we additionally examined the impact of different levels of within-subject variability on decision thresholds. There was a stark difference between the simulated and empirical data when probed through the association between within-subject variability across rounds and decision thresholds. Consistent with our description above, the largest response variability in the Selection bias model was associated with the smallest threshold change across opportunities, and vice versa (r = -0.97, P = 1.70e-40; Fig 7B). It was also the case when part of the variability was attributed to across-opportunity rather than across-round variability in the simulation (Fig 7C). On the contrary, empirical data revealed that

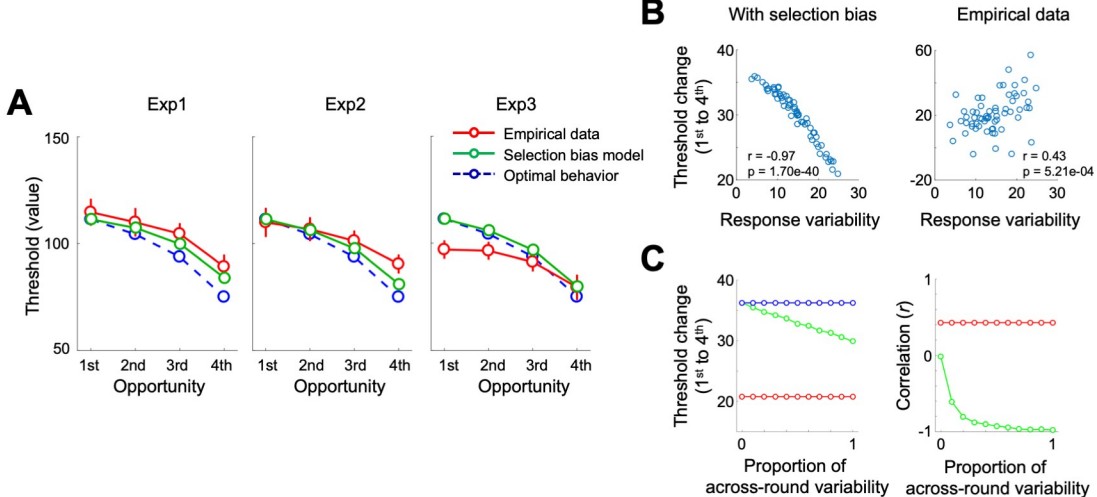

**Fig 7. Subjective optimality model versus a potential impact of selection bias.** Diminishing decision thresholds across opportunities alternatively may be driven by a combination of within-subject decision variability and selection bias where rounds in which decision thresholds happened to be higher (due to the within-subject variability across rounds) lasting further at later opportunities. **(A)** To directly examine this effect, decision thresholds simulated with a selection bias are compared against the optimal decision model and empirical data. The Selection bias model can explain the observed empirical data fairly well for some of the parts in Experiments 1 and 2, but not all of the patterns. Error bars indicate s.e.m. **(B)** We additionally examined the impact of different levels of within-subject variability on decision thresholds. There was a stark difference between the simulated and empirical data when probed through the association between within-subject variability and decision thresholds. The largest response variability in the Selection bias model was associated with the smallest threshold change across opportunities, and vice versa ($r = -0.97$, $P = 1.70e-40$). On the contrary, participants who had the largest response variability showed the largest threshold change across opportunities ($r = 0.43$, $P = 5.21e-04$), which is the exact opposite of the Selection bias model. Each dot represents individual simulation. **(C)** To examine the impact of across-round versus opportunity variability, we ran additional simulations with varying proportion of across-round variability out of the maximum amount of variability. As the proportion of variability attributed to across-round against the across-opportunity variability, the threshold-change gradually decreased from the level of the optimal behavior, and the correlation between threshold-change and response variability asymptotes to -1, decreasing from 0. Levels of threshold change for the empirical data (red) and the Optimal model (blue) are illustrated together as references.

participants who had the largest response variability showed the largest threshold change across opportunities ($r = 0.43$, $P = 5.21e-04$), which is the exact opposite of the Selection bias model. Together, these results suggest that selection bias cannot explain empirical decision patterns observed in this task.

## Discussion

Our results provide a model-based explanation for suboptimality in finite sequential decision-making. Specifically, we present evidence that subjective valuation reflecting individuals' belief about the environment underlies the mechanism of how the brain computes decision thresholds in the problem.

As a classic example of a finite sequential decision problem, various versions of the secretary problem were investigated [1,2]. The standard secretary problem simulates the cases where only the relative ranks matter, such that individuals have to find the best option among the sequentially presented options [7,41]. In this setting, inferior choices lead to no reward, but we have to note that this is hardly the case in real-life. First, any choices we make should have some value even in the case where they were not the best option. For example, an employee who ends up not meeting the employer's original expectation still can make some contribution (unless the employee turns out to be a con artist and shuts down the business). Second, in reality, it is impossible for the decision maker to learn the true relative rank of the chosen option,

because the decision maker will have no knowledge about the subsequent options that were to follow. In other words, there is no one who can examine the success of the choice and deliver a reward if, and only if, the choice were correct. The current study addressed this discrepancy by implementing a task where each option had a monetary reward that matched its face value [4]. Although there was no explicit instruction saying that individuals should find the best option within the finite number of opportunities, participants were informed that the final payoff would be determined by the accumulated reward amount across the entire task and thus, the task preserved the goal of reward maximization (see **Fig J in** S1 File for number of accepted occasions at each opportunity). We believe that the current variation of the secretary problem provides a more naturalistic setting to investigate individuals' sequential decision-making.

A typical behavior pattern observed across various versions of the secretary problem is that individuals show suboptimal choices, such that they wait less than the optimal stopping point [8,17,42]. This suboptimal choice tendency is accounted for by lower decision thresholds than the optimal decision threshold, indicating that they are more likely to accept the option that has low value. However, a recent study reported the opposite pattern, i.e., higher decision thresholds compared to the optimal, in a variant of optimal stopping problem, and suggested a possibility that choice biases may occur in both direction [10]. Among the potential factors why they think they observed the opposite pattern (e.g., usage of naturalistic image stimuli), Furl et al. [10] suspected that a sampling incentive being larger than a sampling cost might underlie participants' motivation to oversample. Although Furl et al. [10] reported that this sampling reward model does not match their empirical data, our results across the three experiments were consistent with this perspective. Specifically, individuals showed higher thresholds for both Experiments 1 and 2 (oversampling), but lower thresholds for Experiment 3 (undersampling). The main change in Experiment 3 was the additional forced wait introduced before the cue when participants were allowed to submit their choice. Our model-based analysis results suggest that this subtle change in task design may have triggered participants to think more about the tradeoff between payoffs and time they spent per round. Such an impact of additional 'cost of waiting (extra time)' is consistent with previous reports showing that non-zero interview cost was associated with lowering decision thresholds [8,17,18,43]. Our model parameter estimates support this interpretation, such that the estimated cost was significantly larger than zero in Experiment 3 where individuals were provided with additional forced waits and in Experiment 2 with more abundant opportunities (K = 10) where individuals were given with further opportunities. These results highlight that the context of decision-making (e.g., task schedule) as well as the extent to which individuals find the task costly (e.g., cognitively demanding or mentally boring) are crucial in decision-making processes [44]. In the current study, we hypothesized a waiting cost as a single component encompassing various types of mental cost and focused on its mechanistic involvement. Future studies exploring the source of the waiting cost may further our understandings about how and why individuals sometimes make choices impulsively or patiently.

Our Subjective optimality model included two free parameters essential in capturing individuals' choice patterns. First, the reference point reflects each individual's belief about the environment [13]. It is known that beliefs can alter how individuals respond to given information [45], and specifically in the current study, we hypothesized that different contexts (i.e., numbers of available opportunities) would change expectations about future outcomes, which in turn would alter individuals' reference points [33]. In line with this, we showed that discouraged expectation (scarce opportunities in Experiment 2; K = 2) causes individuals to be more pessimistic about future chances and wait less in deciding (lowering thresholds). Moreover, encouraged expectation under abundant opportunities (K = 10 in Experiment 2) seemed like causing individuals to be more optimistic about future changes and wait longer. Such an

increment of expectation in response to abundant opportunities parallels a recent study in marketing where they observed higher expectation in individuals' satisfaction for the choices from larger than smaller set of assortments [46]. Interestingly, individuals' expectations (reference point) were significantly higher when they did not have full information about stimuli distribution (Experiment 1). This result suggests that an optimistic bias [47], potentially generated due to the uncertainty about upper bound in the current study, may diminish or even become inverted in other contexts (e.g., scarce opportunities, mental costs).

Second, the nonlinear value sensitivity indicates the extent to which individuals' subjective valuation increases for an additional unit of reward. In the current study, the sensitivity represented as an exponent term in the utility function was smaller than one, which captures marginally diminishing returns for gains and marginally increasing returns for losses [13]. In our suggested model, a range of value sensitivity characterizes a spectrum of decision characteristics in individuals. Value sensitivity close to zero represents a rather categorical valuation (gain or loss relative to the reference point) and choices that are accounted for by a constant threshold being insensitive to the context (i.e., remaining opportunities). On the other hand, value sensitivity close to one represents objective valuation and choices that follow the Optimal decision model. In concert with the reference point, individuals' value sensitivity shapes the extent to which they take into account uncertainty of future opportunities in decision-making. This wide range of individual differences may explain why some individuals are more stubborn with their opinions (e.g., stereotype), while others easily adapt to contextual information [48].

The conclusions of the present study regarding the subjective valuation recruited in determining decision thresholds differ from previous findings by Baumann et al. [34], who argued that people rely on a mental heuristic and determine their thresholds linearly. As illustrated in our formal model comparison (**Fig A in** S1 File), individuals' behavioral choices in each opportunity were explained by the Linear threshold model as well as by the Subjective optimality model. However, the decision patterns between opportunities were not accounted for by a linear decrease in decision thresholds, which is the main feature of the Linear threshold model (Figs 2C, 3D and 4C). This discrepancy suggests that individuals' choice data at least in the current variant of Secretary problem tasks cannot be fully explained by linear heuristic processes. Nevertheless, subtle differences in task designs (e.g., stimuli values followed a normal distribution in Baumann et al. [34], but a uniform distribution in the current study) may recruit different cognitive strategies. On the same line, we acknowledge that there might be other factors (e.g., regret, memory, selection bias) and their combination which would be able to explain choice patterns during the task, and thus future studies should explore potential factors that affect individuals' decision processes.

In the current study, the pupil responses encode both decision difficulty and the subsequent choice of whether individuals will accept or reject the presented stimulus. Both types of information temporally preceded actual choice, so these pupil dilations are the physiological representations of the processed information regarding decision-making, rather than a simple reflection of the presented visual information [26]. As suggested from previous studies, pupil dilation may reflect the downstream processing of the anterior cingulate cortex [24,25], the brain region that is involved in encoding decision difficulty [49], and, more broadly, a wealth of value-related information—including difficulty signals—during decision-making processes [40]. Differential pupil sizes depending on the subsequent choices suggest that there is more to neurophysiological representation than simple decision difficulties. The pupil responses may be reflecting the involvement of the neuromodulatory systems (e.g., locus coeruleus [50]) in decision processes through release of norepinephrine, which temporarily facilitates computations in cortical networks [22,26]. During the process of decision-making, individuals who had higher value sensitivity showed exaggerated pupil responses reflecting both decision

difficulty and subsequent choice information. This correlation result indicates that 'behavior-level sensitivity' we extracted from our model is consistent with an independent 'neurophysiology-level sensitivity'. Consistent with previous studies, our pupillometry results evidence the association between neurophysiological pupil responses and individuals' characteristics in valuation [27], and support a mechanistic account of seemingly suboptimal decision strategies in finite sequential decisions.

Alternatively, the observed pupil responses could be a result of top-down control, such that individuals may pay more attention to the stimuli that they plan to accept for accumulating more evidence [51] or a process following the opposite causality, in that the rich amount of accumulated evidence of a particular stimulus may induce even higher attention levels (e.g., saliency driven bottom-up attention [52]). In the current study, the latter is unlikely, given that all low-level visual information (e.g., contrast) of the displayed stimuli were matched or controlled for. To sum up, the current results show that the two pieces of information essential in subjective valuation are linked together at the physiological level, deeply involved in the process of decision formation [26].

The future direction of the current study includes expanding our model further, so that it can capture more realistic decision contexts where various factors may interact (e.g., confidence, concerns about sunk cost, limitation of information processing capacity). The first and foremost extension should be to explain the mechanisms of how individuals learn the stimulus distribution (e.g., reinforcement learning). Previous studies reported no evidence of learning in various versions of the secretary problem [8,53]. In line with these studies, in the current study, we assumed that the learning process is rapid and negligible in relevance to other decision processes. Nevertheless, we conducted two explorative analyses examining potential learning effects. First, each individual's task data were divided into the first and the second half. Across all three experiments, decision thresholds estimated from the first and the second halves were comparable, indicating no sign of extensive learning (**Fig H in** S1 File). Second, we ran a series of logistic regressions to examine the impact of presented stimulus at the preceding opportunity in subsequent choices. There was a slight hint of negative influence of preceding stimuli on subsequent choices, but the beta coefficients were less than 1/10 of the betas of the current stimulus (**Fig I in** S1 File). Moreover, we showed that decision processes under imperfect information (no knowledge of the maximum value) were comparable with the processes under the full information. These results suggest that, even without explicit information about the stimuli distribution, people, in general, have a rough idea about the range of values of an uncertain option. Alternatively, people were able to learn early enough [6] that the behavioral strategy for the rest of the task was not different from the case where individuals knew about the distribution from the beginning (c.f., see [35] for dynamic reference point). Still, inclusion of learning mechanisms in the model (e.g., learning what to expect, regretting previous choices) would be essential to examine whether or not the decision model generalizes to broader contexts (e.g., volatile environment).

Examples of finite sequential decision problems span a wide range of life choices, including finding the right life partner and choosing a career, the aims of which are to maximize reward under a limited amount of resources and opportunities. Such value-based decision processes with reference to costs are not unique to humans but extend from fish choosing a mate, who become less selective under costly environments [54], to primates making foraging decisions [55]. The Subjective optimality model provides a way in which individual subjective valuation generates systematic biases in sequential decision-making and opens a window to decompose physiological responses into decision difficulty and signatures of subsequent choice, of which levels differ in the extent of individual value sensitivity. In sum, our data support a mechanistic account of suboptimal choices varying from overly impulsive

choices in individuals with substance-use problems [56] to delayed choices in individuals who suffer from indecisiveness [57].

## Methods

### Ethics statement

The current study was approved by the Institutional Review Board of Ulsan National Institution of Science and Technology (UNISTIRB-18-39-C, UNISTIRB-18-14-A). All participants provided written informed consent and were paid for their participation.

### Participants

One hundred eleven healthy young adults (male/female = 55/56, age = 22.72 ± 2.04 years) participated in the current study. None of the participants reported a history of neurological or psychiatric illness. Three separate experiments were conducted and there were no overlapping participants across experiments. Twenty students participated in Experiment 1 (male/female = 10/10, age = 22.85 ± 1.31 years), and 67 students were recruited for Experiment 2 where they had two, five, or ten opportunities per round (male/female = 33/34, age = 22.70 ± 2.15 years). Among the participants in Experiment 2, three participants were excluded from the analyses due to their reported suspicion about the payment structure of the experiment. Among the included participants, 23 participants (male/female = 11/12, age = 23.09 ± 2.09 years) were assigned to the condition where they were given two opportunities per round, 21 students (male/female = 11/10, age = 23.19 ± 1.86 years) were randomly assigned to the condition where they were given five opportunities per round, and 20 participants (male/female = 10/10, age = 22.00 ± 2.41 years) were assigned to the condition where they were given ten opportunities per round. Twenty-four students participated in Experiment 3 (male/female = 12/12, age = 22.67 ± 2.28 years). Two participants were excluded due to their reported suspicion about the payment structure of the experiment, and one participant was excluded due to data loss from a computer error. Three participants were excluded from the pupil diameter analyses due to poor calibration. After exclusion, data from 21 participants (male/female = 11/10, age = 22.62 ± 2.38 years) were used for behavioral analyses, and a subsample of the data (N = 18; male/female = 8/10, age = 22.33 ± 2.30 years) was used for further pupil diameter analyses. All participants reported normal or corrected-to-normal vision under soft contact lenses (no glasses were allowed due to potential reflections during eye-tracking). The sample size of each experiment was based on those used in similar studies using finite sequential decision-making paradigms [e.g., 7,17].

### Stimuli and apparatus

All stimuli were generated using Psychophysics Toolbox Version 3 (http://www.psychtoolbox. org/) and MATLAB R2017a (MathWorks), and presented on a DLP projector (PROPixx VPX-PRO-5050B; screen size of $163 \times 92$ cm$^2$; resolution of $1920 \times 1080$ pixels; refresh rate of 120 Hz; linear gamma). The distance between the participants' eyes and screen was fixed at 153 cm. The ambient and background luminance were set at 1.1 and 69.2 cd/m$^2$, respectively. The main stimuli were three-digit integer numbers, randomly selected between zero and 150. To minimize luminance effects on pupil size, one- or two- digit numbers were displayed as three-digit numbers with extra zeros attached in front of the stimuli (e.g., 1 is displayed as '001'). During the task, fixation was enforced at the center of the screen with an infrared eye tracker (Eyelink 1000 Plus, SR Research, Canada), and a chin and forehead rest were used to minimize head movement.

## Experiments

At the beginning of the task, the eye-tracker was calibrated, referencing eye fixation data at the four corners of the screen. During the task, participants made a series of choices either to accept or to reject presented stimuli (Fig 1A). As explained above, the stimuli were randomly selected integers between zero and 150 where each number had equal probability of being selected (uniform distribution). Each presented number could be considered as an option whose value matches its face value because participants were instructed that all accepted numbers would be added to their final payoff at the end of the task. Given this knowledge, participants had a fixed number of opportunities (chances) to evaluate and reject a new randomly selected number. The present 'round' ended when participants accepted a presented number within this limited number of opportunities, or when they ran out of the opportunities where they had no other choice but to accept the presented number at the last chance. At the beginning of each opportunity, participants were shown which opportunity they were currently at, so that they would not lose track of the number of remaining opportunities. A new round followed, at which the number of available opportunities was reset to the original maximum quantity. Participants were paid at the end of the study (after completing 200 rounds), based on the sum of the numbers they chose during the task. All instructions were provided through illustrated slides.

Overall, we implemented three separate experiments, each of which had slightly different settings. In Experiment 1, participants had up to five opportunities (K = 5), and they were not explicitly informed of the maximum number (150) that would be presented. Use of the context with incomplete information was to incorporate a more naturalistic setting as real-life problems where, as in most of the cases, individuals do not have knowledge about the best potential option (e.g., even if the current candidate for a job has a good enough fit for the position, one cannot assure that a potential future candidate will not have a superior fit). Participants were instructed that the presented stimuli would be sampled from a uniform distribution, and thus, we expected that participants would quickly deduce the maximum range through iterative experiences. At the beginning of the new round, the accumulated payoff amount up until the last round was presented at the bottom of the screen. In Experiment 2, participants were randomly assigned to one of three conditions where one condition had two (K = 2), one condition had five (K = 5), and one condition had ten (K = 10) opportunities. Here, participants were also informed of the maximum number (i.e., 150). In addition, participants were given a practice session that comprised two rounds where all the stimuli were '000', which allowed them to be familiarized with associated buttons and the task screen settings. All the rest of the task settings were identical to Experiment 1.

Experiment 3 was designed to temporally dissociate actions (i.e., accept or reject) from the stimulus onset, so that physiological responses to stimuli independent from potential motor preparatory signals could be measured. Particularly in Experiment 3, participants were not allowed to make choices until an audio cue was played (Fig 4). The audio cue was played between 1.5 and 2.5 seconds after stimulus onset (uniform distribution), which allowed us to tease out potential confounding factors related to action from the pupil diameter measures at 0–1.5 seconds after stimulus onset. In addition, to prevent participants from making unnecessary eye movements, all the information including number stimuli were presented at the center of the screen. As implemented in Experiment 2 where K = 5, participants were informed that the maximum number was 150 and that they have up to five chances to evaluate the stimuli per each round.

### Behavioral analysis

For all three tasks, behavioral choices (accept or reject) and response time (RT) were measured. Individuals' decision threshold for each opportunity was estimated from their choices. To estimate empirical decision threshold for each opportunity, a cumulative distribution function of Gaussian distribution was fitted to individuals' choice data that corresponded to the same opportunity across all 200 rounds. The mean and variance parameters of the Gaussian distribution represent the decision threshold and decision variability, respectively. A set of best-fitting parameters that maximize the likelihood of the data was estimated per individual using the Nelder-Mead simplex algorithm provided by MATLAB R2017b.

### Computational modeling and model comparison

For a formal model comparison at the group level, choices from all 200 rounds per participant were used for parameter estimation. We used likelihood-ratio tests to compare goodness-of-fit of the models for explaining participants' decisions.

### Optimal decision model

An optimal decision maker is expected to maximize their payoff by estimating the expected value of each opportunity. This computation can be conducted from the final opportunity to the first, given the full information about the stimuli distribution (U[0, 150]). For example, in a condition where K = 5, the expected value of the last opportunity is 75, and therefore a payoff-maximizing optimal decision maker should set 75 as the decision threshold of the fourth opportunity (i.e., accept numbers larger than 75 and reject those that are lower). Then, this decision strategy should again determine the expected value of the fourth opportunity. Generalizing this dynamic programming approach, the decision threshold of the i$^{\text{th}}$ opportunity ($\vartheta[i]$) can be written as follows:

$$\vartheta[i] = \frac{\lfloor \vartheta[i+1] \rfloor}{151} \vartheta[i+1] + \frac{1}{151} \sum\nolimits_{v=\lfloor \vartheta[i+1] \rfloor + 1}^{150} v \qquad (i \in (K-1, K-2, \ldots, 1))$$
$$\vartheta[K] = 0$$

where $\lfloor x \rfloor$ indicates the greatest integer less than or equal to x.

### Subjective optimality model

Our hypothesis was that individuals use subjective valuation in reference to their own expectations about the environment during finite sequential decision-making. To test the hypothesis, we constructed a computational model drawn upon Prospect theory [13]. Particularly, individuals' subjective valuation (Util) of an objective value (v) was defined as below:

$$\text{Util}(v) = (v - r)^\rho \text{ if } v \geq r$$

$$\text{Util}(v) = -(r - v)^\rho \text{ otherwise}$$

where $\rho$ and r indicate individuals' nonlinear value sensitivity and reference point, respectively. Subjective valuation is also used in computing decision thresholds:

$$\vartheta[i] = \text{Util}^{-1} \left( \frac{\lfloor \vartheta[i+1] \rfloor}{151} \text{Util}(\vartheta[i+1]) + \frac{1}{151} \sum\nolimits_{v=\lfloor \vartheta[i+1] \rfloor + 1}^{150} \text{Util}(v) \right)$$

where Util$^{-1}$(.) indicates an inverse function of the aforementioned subjective value function and Util($\vartheta$[i+1]) indicates the expected utility of continuing the game.

### Subjective optimality model with a waiting cost

In our secretary problem task, choosing to reject the current stimulus means that participants have to go through further steps (opportunities) to receive rewards (or at least to find out how much reward they will receive) until they choose to accept at a later opportunity. Such an additional wait may introduce a disutility (i.e., negative value) against the choice to reject. To test this possibility and quantitatively estimate this 'mental waiting cost', we modified our suggested Subjective optimality model to a more general format as follows:

$$\vartheta[i] = \mathrm{Util}^{-1}\left(\frac{\lfloor\vartheta[i+1]\rfloor}{151}\mathrm{Util}(\vartheta[i+1]) + \frac{1}{151}\sum\nolimits_{v=\lfloor\vartheta[i+1]\rfloor+1}^{150}\mathrm{Util}(v) - C\right)$$

where C indicates a waiting cost per opportunity. Note that the waiting cost lowers the expected utility of the following opportunity ($i+1^{\mathrm{th}}$), and thus has an effect of lowering the decision threshold of the current opportunity ($i^{\mathrm{th}}$).

### Constant threshold model

There is a simple alternative decision strategy for the secretary problem: to use a constant decision threshold throughout all opportunities. To examine this possibility, we estimated one decision threshold per individual. This constant threshold model provides a quantitative baseline for a formal model comparison.

### Independent threshold model

The Independent threshold model is the model in which a decision threshold for each opportunity was independently estimated. This model does not hypothesize any mechanistic associations among opportunities, but focuses on capturing individuals' choice tendencies in each opportunity.

### Linear threshold model

In a recent study, Baumann et al. [34] suggested a linear threshold model that assumes a linear relationship among the thresholds across the opportunities (i.e., Threshold$_{\mathrm{i\text{-}th}}$ = a + bi where a and b are free parameters) [34]. Note that in this Linear threshold model, a decision maker does not need to use dynamic programming approach. One key feature that clearly dissociates the Linear threshold model from other models is the linear relationship in decision thresholds between opportunities.

### Regret model

We constructed a model where participants would accept candidates that are higher than both the optimal threshold and the highest value they have passed up in previous opportunities. To examine the maximum impact of past stimuli values, we applied the aforementioned rule of regret as deterministic decision thresholds at each opportunity and also assumed perfect memory. That is, all the past stimuli values, rather than the most recent stimulus, affect decisions within the corresponding round. We simulated 10000 rounds of behavioral choices of pseudo-subjects, and estimated the decision thresholds and the effects of past values on the current decision. These steps are repeated 20 times to verify the reliability of the estimations. Because the model is a combination between the Optimal decision model and the regret, decision thresholds at the initial opportunities are always the same as those from the Optimal decision model. Note that in this simulation-based analysis, we are using the differences in slopes across decision thresholds, rather than other features (e.g., the deviation at the initial opportunities in

Experiment 3), as criteria to examine whether the simulation model captures the core mechanism explaining the empirical data.

## Selection bias model

One of the sources of potential variability in individuals' decision thresholds is within-subject variability that affects individuals' across-round behaviors. To run the simulation, we calculated the behavioral choice variability from the entire data (across- opportunities and rounds), and assumed it all would contribute to the within-subject variability across rounds (i.e., there is no across-opportunity variability within a round). This is the most conservative setting, because large across-round variability opens up the possibility that only the rounds in which decision thresholds happened to be high decision thresholds to contribute to the post-hoc estimation of the later opportunities; in turn, this would make the estimated decision thresholds to have a shallower slope (more flat) across opportunities compared to the optimal decision model.

## Predicting change of decision threshold based on the altered decision context

To examine whether or not our suggested model can be generalized under different contexts with scarce opportunities, we took a prediction approach using model-based information from the context with abundant opportunities. Specifically, the reference point and nonlinear value sensitivity parameters estimated from behavioral choices of individuals (N = 21) who participated in Experiment 2, K = 5 were used to predict the decision threshold in the two (K = 2) and ten (K = 10) opportunities condition. Particularly for the nonlinear value sensitivity, the parameter distribution in the K = 2 or 10 condition was assumed to be the same as that in the K = 5 condition. On the other hand, the parameter distribution of the reference point was assumed to be shifted down by the difference of expected earnings between the two conditions, reflecting participants' acknowledgement of the scarce or abundant number of opportunities. Specifically, we calculated the expected value across all opportunities for each condition (K = 2, 5, and 10), and assumed their difference to indicate the extent to which participants changed their expectation about earnings from average number of opportunities (K = 5) to either abundant (K = 10) or scarce (K = 2) opportunities. To predict the mean threshold in the K = 2 condition, 23 pairs of parameters (matching the number of participants in K = 2) were randomly sampled with replacement from the aforementioned parameter distribution, and the thresholds corresponding to each parameter pair were computed by applying our model. The mean threshold in the K = 10 condition was predicted following the same procedure. The procedure was repeated 5,000 times to estimate the distribution of the mean of 23 thresholds. The 95% confidence interval was computed from the 5,000 means.

## Parameter estimation procedure

We used Bayesian hierarchical analysis to estimate the best-fitting parameters for participants' choice data [58]. Here in all the tested models, we introduced a gaussian noise around computed decision thresholds, which mirrors the stochastic nature of choices rather than assuming that choices are made deterministically before and after each threshold value (as applied in Signal detection theory). The width of the gaussian noise is set as a free parameter (termed as a decision variability s) and estimated in individual-level.

The parameters characterizing individual participants were drawn from the population distributions, each of which follows a Gaussian distribution. The priors on the means of the population distributions ($\mu$) were set to broad uniform distributions, and the priors on the SDs ($\sigma$)

were set to an inverse-Gamma distribution in each of which, the shape parameter alpha is one and the scale parameter beta is manually selected. To improve sampling efficiency, we sampled the parameters from a transformed space, and the hierarchical structure was assumed in the transformed space. Specifically, the reference point and value sensitivity parameters were sampled without domain restrictions and transformed by a scaled logistic function $g(x) = A/(1 +exp(-x))$ before applying to the model. In the function $g(x)$, A was set to 150 for the reference point parameter $r$, and set to 2 for the values sensitivity parameter $\rho$. The decision variability parameters and the group-level hyper-parameters for parameters' standard deviation were transformed by $exp(.)$ after sampling. We did not apply a transformation to the waiting cost parameter. A Markov chain Monte Carlo (MCMC) method (Metropolis-Hastings algorithm) was used to sample from the posterior density of the parameters conditioned on all of the participants' choices. We estimated the most likely set of parameters for each participant from the resulting chain of samples using a multivariate Gaussian kernel function provided by MATLAB R2017b.

## Pupillometry: Preprocessing

Pupil diameter was sampled at 500 Hz from both eyes using an infrared eye-tracker (Eyelink 1000 Plus; SR Research, Kanata, Canada) and recorded continuously for the entire session. Blinks and saccades in each eye were identified using the standard criteria provided by Eyelink, and the identified intervals were linearly interpolated. Particularly for the blink events, the interpolation was applied to the intervals between 150 ms before and after each identified blink. Three participants whose pupil data included a large proportion of interpolated intervals ($> 50\%$) were excluded from further analyses. The means of the interpolated data from both eyes were band-pass filtered between 0.02–4 Hz using third-order Butterworth filters. The long-lasting effects ($\sim 5$ sec) of blinks on pupil diameter were identified by applying least-squares deconvolution to individual data, and then removed from the data [59]. Then, the resulting data were z-scored for each session (i.e., each participant). Pupil diameter changes in response to the value stimulus were computed for each opportunity. Each epoch was defined for pupil responses between -200 and 1,500 msec around the stimulus onset, and corrected for its baseline by subtracting the mean pupil size around ($\pm 20$ msec) the onset. The choice trials that required a large proportion ($> 50\%$) of interpolation were excluded from the analysis, which comprised 28% of the entire choice trials. Applying more liberal exclusion criteria (excluding the trials that required substantially large proportion ($>90\%$) to be interpolated) did not alter any of the pupillometry results (**Fig F in** S1 File).

## Pupillometry: Statistical tests

To examine whether physiological responses reflect cognitive decision processes, we tested pupil dilations and contractions in response to (i) subsequent choices to accept or reject, and (ii) decision difficulty. First, pupil diameter changes between 0–1,500 msec after the stimulus onset were compared between accepted and rejected opportunities. We used t-tests to compare mean differences at each time step and defined statistically significant temporal clusters (alpha level set to 0.05). To control for the false alarm rate, we used the cluster-based permutation method [38] and examined the statistical significance of each cluster. Particularly in the permutation procedure, the sign of the difference value for each participant was randomized and the sum of t-values in each cluster was used as its statistic. Second, the peak pupil dilation between the stimuli onset and 1,500 msec after the stimulus onset was used to examine the effect of decision difficulty—the absolute distance between the corresponding decision threshold and the presented value—on the pupil dilation. Linear regression was used for the rejected

trials (choice = reject, -40 < Value—Threshold$_{i\text{-th}}$ < 5) and accepted trials (choice = accept, -5 < Value—Threshold$_{i\text{-th}}$ < 40) separately for each participant. The same set of data points was used to test the effect of choice on pupil dilation after controlling for the decision difficulty. We further investigated individual differences in the extent to which one responds to stimulus value at the physiological level (i.e., pupil dilation). We smoothed each individual's pupil dilation data along the threshold centered values (i.e., Value–Threshold$_{i\text{-th}}$) ranging from -90 to 60 by applying local regression using a 2D polynomial model provided by MATLAB R2017b. The estimated pupil dilation at threshold was used to calculate the Pearson correlation between individuals' estimated value sensitivity and their pupil responses.

## Supporting information

**S1 File. Supporting information.** A .docx file containing supporting text and figures. (DOCX)

## Acknowledgments

We thank Youngyoon Kim, Hyeji Lee, and Jeong Yeo for their research support.

## Author Contributions

**Conceptualization:** Yun Kyoung Shin, Oh-Sang Kwon, Dongil Chung.

**Data curation:** Yeonju Sin, HeeYoung Seon.

**Formal analysis:** Yeonju Sin, HeeYoung Seon, Oh-Sang Kwon, Dongil Chung.

**Funding acquisition:** Oh-Sang Kwon, Dongil Chung.

**Investigation:** Yeonju Sin, HeeYoung Seon, Oh-Sang Kwon, Dongil Chung.

**Writing – original draft:** HeeYoung Seon, Oh-Sang Kwon, Dongil Chung.

**Writing – review & editing:** Yeonju Sin, HeeYoung Seon, Yun Kyoung Shin, Oh-Sang Kwon, Dongil Chung.

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
