## [Decision Letter · Decision Letter 0]

6 Oct 2020

Dear Dr Chung,

Thank you very much for submitting your manuscript "Subjective optimality in finite sequential decision-making" for consideration at PLOS Computational Biology.

As with all papers reviewed by the journal, your manuscript was reviewed by members of the editorial board and by several independent reviewers. The reviewers raised a significant number of concerns about the paper. Some of these concerns relate to the modeling framework, with reviewers suggesting the paper does not go far enough in considering alternative models and that the appeal to prospect theory might be a more complex model than can be justified. The reviewers also noted shortcomings in the discussion of relevant prior literature, as well as concerns about aspects of the data analysis, including possible selection biases and learning-related effects. Finally, all reviewers noted that the pupil dilation results are somewhat disconnected from the main questions that the paper considers, at least in the way that the paper is currently framed.

Although a substantial number of concerns were raised, we would like to invite the resubmission of a significantly-revised version that takes into account the reviewers' comments.

We cannot make any decision about publication until we have seen the revised manuscript and your response to the reviewers' comments. Your revised manuscript is also likely to be sent to reviewers for further evaluation.

Sincerely,

Adrian M Haith

Associate Editor

PLOS Computational Biology

Samuel Gershman

Deputy Editor

PLOS Computational Biology

Reviewer's Responses to Questions

**Comments to the Authors:**

Reviewer #1: This is an interesting paper looking at decision making in sequential search problems, a.k.a. secretary problems. The authors hypothesize that rather than using an optimal value maximizing strategy, participants instead have a reference point and behave as if they are maximizing utility with respect to that reference point. They test one prediction of this in Experiment 2, by showing that people appear to have higher reference points, and thus thresholds, when there are more opportunities. In Experiment 3 they also collect pupil dilation data and study its relation to choice outcomes and difficulty. They show that pupil dilation increases with difficulty, but is also higher when accepting opportunities.

Overall, I think that this is an interesting topic that gets a lot of attention in economics and business journals, but could benefit from a neuroscience perspective. I also think the authors are likely onto something with the Prospect Theory story (but see below). My enthusiasm is dampened by a few issues, some which could be addressed by expanding the scope of the literature review, others which would require new analyses and could change the takeaway messages of the paper. My view is that there are a collection of (potentially) interesting results here, but they don’t all hang together very well, and they don’t quite tell a compelling, coherent story.

One potential major problem that I see is an issue of selection bias in the analyses. There will always be some subject-level heterogeneity in behavior, in this case in the thresholds that participants use. Subjects with lower threshold strategies will tend to accept earlier opportunities and so be underrepresented at subsequent opportunities. This will make it look like the thresholds are relatively flat across opportunities, compared to the reality. This is particularly an issue for Experiment 2, where the authors compare behavior in Opportunity 1 (out of 2) to Opportunity 4 (out of 5). The authors find that participants in the former case have a lower threshold than those in the latter, which seems like an important finding, and provides crucial support to their Prospect Theory explanation. However, it could simply be that all the “low threshold” participants have disappeared by Opportunity 4, while the entire population is present in the Opportunity 1 analysis. The authors could try to address this by including trials where participants accepted prior to Opportunity 4, and estimating what their threshold would have been in Opportunity 4, based on what they accepted earlier. This would seem to require some extrapolation, so I’m not sure this analysis would ever be clear cut. The authors might want to look at the Heckman correction for missing observations.

The previous point notwithstanding, I have to question the evidence for the Prospect Theory (PT) explanation. While PT is well established and offers a sensible explanation here, I wonder what it adds empirically? How does this model with a reference point do more than just fitting a threshold model where the initial threshold and threshold decay rate (or time cost) vary across participants? Where is the model comparison with Optimal+C for instance? I guess I’m wondering whether the PT model yields predictions that can be falsified or whether it is so flexible that it could match just about any pattern of results? This is a crucial part of the paper, and I feel like the evidence for it is lacking.

The pupil dilation analysis (Experiment 3) is a very important part of the paper, but it feels somewhat tacked on and disconnected from the rest of the story. This is in part due to the fact that the authors never mention pupil dilation in the introduction of the paper. They do briefly discuss it prior to Experiment 3, but their literature review is a single sentence, and their hypotheses are very vague, stating only that “pupil diameter may capture value of stimuli, decision difficulties, and final choices…” What exactly did the authors hope to see here? How do these pupil results contribute to our understanding of pupil dilation, or of the secretary problem?

Finally, on a related point, I found the connections to the literature somewhat lacking. There has been a lot of work on pupil dilation by people like Jonathan Cohen, Yale Niv, Robert Wilson, Kerstin Preuschoff, Eran Eldar, Joshua Gold, etc. The authors could do a much better job situating their results in that literature. More importantly, there has been quite a lot of work on consumer search problems in economics and business, yet there are only a handful of references here (only 3 in the introduction!). A quick Google Scholar search yields many results for the combination “consumer search prospect theory”, with relevant papers by e.g. Kogut, 1990. Prospect Theory is so pervasive in JDM research that there must be more related papers here.

Other comments:

Ratcliff 1978 is not a great reference for easier decisions being faster. That is a paper about memory retrieval. The authors might consider references on value-based decision making, e.g. Mosteller & Nogee 1951; Clithero 2018; Konovalov & Krajbich 2019.

It might be helpful to display something like violin plots of accepted numbers for each of the opportunities. Is the distribution really Gaussian, or close to it?

Doesn’t the declining threshold result rule out a flat threshold strategy? How does Fig. 1c look if you use the optimal thresholds instead?

“till” —> until

It seems worth noting that if time is costly, you would expect even lower thresholds than “optimal”, not higher. This is true in Experiment 3, but not Experiment 1 or 2.

Why compare models with likelihood ratio and not something like AIC/BIC/DIC/WAIC? I also was unclear exactly why the chi-square test has ~20-60 degrees of freedom. I assume this is because each subject has their own parameter value. What if instead you modeled it like a random-effects model, where there is a population-level mean and variance for the parameter? Then there would only be 2 degrees of freedom. On a related note, Table S1 could do a better job indicating which model wins the comparison.

In the model, I take it that U(theta[i+1]) means the expected utility of continuing the game, not the expected utility of accepting that amount? This could be made clearer.

Fig.2B is missing the optimal behavior.

Did the authors really hypothesize that the wait time would increase the cost of waiting and thus lower thresholds, or was the wait time just something they did for the pupil measurements?

Shouldn’t the reference point decrease if you know you’re going to be more impulsive and choose earlier?

Is the peak in pupil dilation near threshold simply due to the fact that these decisions are taking longer, and pupil dilation continues to rise with time? To help address this question, it would be helpful to see a version of Fig. 6a (or Fig. S2a) where the curves are broken down into |value-threshold| bins. This continuous rise in pupil dilation over time is well documented in the literature, include here (Fig. 6a). Along those lines, I wonder how Fig. 6B would look if the authors used something like “peak pupil dilation” rather than a specific time like 1500 ms?

“humans in general have optimistic bias” - I disagree with this interpretation of the results. A simpler explanation is that participants simply don’t know the upper bound, but do know that the lower bound is zero. This will naturally cause an “optimism bias”, without any actual optimism.

What was the suspicion about the payment structure that some participants expressed?

P.31 bottom - “the change of participants’ expectation about mean earning was calculated comparing expected values between conditions…” What exactly does this mean? Using the optimal thresholds?

Excluding 28% of trials from the pupil-dilation analysis is a lot. Are the results robust to less extreme exclusions?

The authors mention some pupil-dilation smoothing. They mention “-90 to 60”. What exactly are these numbers? What are the units?

Reviewer #2: Summary:

In this paper, Shin et al. examine human choice behavior in a sequential decision problem that is a variant of the classic “secretary problem.” In the problem examined, “candidates” are drawn from a uniform distribution between 0 and 150, and on each round, the chooser decides whether to accept the current candidate or reject it and move on to the next round. Choosers face a total of 200 problems and are paid proportional to the sum of all candidates selected. Across experiments, whether choosers are told about the distribution or not is varied, as is the number of rounds (either two or five rounds are used). Under the assumption that the chooser knows the full distribution of candidates, the optimal solution can be easily obtained using dynamic programming, and takes the form that the chooser accepts any candidate above a certain decision criterion and that the optimal decision criterion only depends on the number of remaining candidates. The primary empirical result, replicated across three experiments, is that human choosers do not adjust their decision criterion as strongly as the optimal solution, with a resulting bias towards higher or lower criteria in different experiments. The authors account for choice behavior across experiments assuming a dynamic programming-like model in which, rather than actual values, candidate values are evaluated in a prospect theory-like manner with a non-linear subjective function relative to a reference point. Measurements of pupil diameter in a third experiment also show that pupil diameter is influence by choices and choice difficulty.

Comments:

1. Though the empirical result is a novel contribution to the literature, the modeling is less compelling. The authors assume that participants solve the problem in a dynamic programming-like manner, but they do not do enough to establish this basic assumption. They do not adequately explore alternative possibilities enough to make such a claim. The only alternative model that is considered is a fixed decision criterion model. Several other critical possibilities are not explored, including (a) that participants’ choices were dependent on the candidates they have passed on in the current round and (b) that participants learn over problems how to adjust their decision criterion in order to maximize outcomes. Ruling out these alternatives is particularly important as there are reasons to doubt that human choosers use the kind of backward induction approach proposed here (eg, many examples in game theory where choosers only look a small number of steps forward/backward).

2. The motivation for a prospect theory-like evaluation is not entirely clear. Is the reference point meant to be the expected outcome from the problem? If so, why wouldn’t that reference point change over problems as participants learn to better estimate the expected outcomes? And why wouldn’t the reference point change as a function of the remaining number of potential candidates? The assumption that the reference point is constant across rounds and problems is not convincing. An important feature of prospect theory is that the reference point can flexibly change depending on the decision context.

3. The model explains the difference between Exp 1 (where the distribution is unknown) and Exp 2 (where the distribution is known) by fitting a higher reference point to Exp 1 data. However the difference between these experiments is a critical place where potential learning effects should be examined. As participants experience more rounds, they should build more accurate beliefs about the candidate’s distribution, which should shift the reference point in the model. The authors assumed the reference point was constant throughout the experiment, but it’d be reasonable to test this assumption by comparing reference points between the 1st half and the 2nd half.

4. The model explains the difference between Exps 1 and 2 (where decision criteria are too high in later rounds) and Exp 3 (where decision criteria are too low in early rounds) by including a “waiting cost” parameter that is only non-zero in Exp 3. This modeling solution feels ad hoc. How much time do the participants save (relative to a waiting cost of zero) and how much payment do they give up to save it?

5. The pupil data largely replicate previous results showing an influence of choice and choice difficulty on pupil. That is, these data do not seem to be the major contribution of the paper, above and beyond the behavioral data and (potentially) modeling. The one novel finding here is an association between pupil dilation and value sensitivity. However, the authors write “This positive correlation between value sensitivity and pupil dilation was statistically significant at the threshold where decision difficulty is the highest” (p. 19, line 371-372). Were multiple comparisons corrected for in this analysis, and if so, how?

6. This is a minor point, but the decision problem and what participants actually know about the problem in each experiment needs to be explained more clearly at the outset of the results section. (Eg, do they know it’s a uniform distribution, do they know the max of the distribution, do they know the number of rounds, do they know the number of problems, do they know the payoff function, etc). All of this information can be gleaned from the methods, but this section is at the end of the manuscript, and this information is critical to understanding the modeling and the differences across experiments.

Reviewer #3: This paper developed a computational model of subjective valuation to explain suboptimal behavior in finite sequential decision making contexts. The authors were interested in testing and justifying their model, including a physiological basis for their explanation. I commend the authors for pursuing this timely and important question.

Nevertheless, there is room for significant improvement in clarifying the authors’ claims. While the experimental results are compelling, the authors should establish much better context for their claims. Secondly, this affects the extent to which they can generalize their results. These concerns are outlined below, and I hope the authors can use this feedback to improve their manuscript.

Major Comments

1) The introduction does not establish sufficient context for their experiments. It is unclear a) What other explanations/models there are for the secretary problem, b) Why subjective valuation may be a good model to explain this behavior, c) What link pupillometry has to subjective value?

2) Given the experimental design, K is a critical independent variable and it is unclear how the experimenters defined it at K=2 and K=5. What happens if K is increased to 10, 20, or 100? At minimum, K needs to be explained and justified more completely. The issue is that it is currently unclear how the results between Experiment 1, 2, and 3 build on each other. Especially going from experiment 2 to 3, it remains unclear whether the results are due to careful manipulations, or rather due to unintentional confounds introduced by virtue of changing the task design. Additional behavioral data could go a long way toward strengthening the authors’ claims here and improving the manuscript, but it may alternatively simply require expanded rationale and justification in the main text.

3) The authors claim that Experiment 1 and 2 are in line with affecting neural hypotheses, whereas in fact these are purely behavioral results. This should be developed further in the Introduction and in the description of the results.

Minor Comments

1) An alternative explanation for the pupillometry results is that these systems are acting upon stress/norepinephrine networks. Notably, some studies have indicated that fluctuations in pupil diameter are associated with BOLD changes in the locus coeruleus (e.g., Murphy et al., 2014, HBM).

2) Among other limitations to external validity in the secretary problem include that these designs fail to constrain a decision-makers opportunity costs of time and computational capacity. As a result, it is confounded whether a decision-maker is suboptimal in a more realistic setting or is in fact optimizing factors that are more important to them (e.g., undersampling a pool of candidates because it requires less work, so that the team leader can focus on a project deadline). This issue could be considered more thoroughly in the Discussion section.

3) Related to the point about the Introduction, please establish a link between behavioral value sensitivity and subjective value and the relationship of pupillometry to these concepts.

4) It is not clearly justified how decision difficulty and evidence accumulation are associated with the subjective value model. How would response time factor into these models and the behavior? Could an alternative model employ the drift diffusion model to assess changes in valuation (e.g., Hutcherson et al., 2015, Neuron)?

5) The authors should explain why their experimental design is inconsistent with many results in the secretary problem, where other researchers have found that participants tend to undersample a pool of information.

6) The GitHub link for the code/data is not working. Please check.

**Have all data underlying the figures and results presented in the manuscript been provided?**

Reviewer #1: Yes

Reviewer #2: Yes

Reviewer #3: **No: **The GitHub link isn't working, which should be easily remedied in a revision.

PLOS authors have the option to publish the peer review history of their article (what does this mean?). If published, this will include your full peer review and any attached files.

Reviewer #1: No

Reviewer #2: No

Reviewer #3: No
---

## [Decision Letter · Decision Letter 1]

13 May 2021

Dear Dr Chung,

Thank you very much for submitting your manuscript "Subjective optimality in finite sequential decision-making" for consideration at PLOS Computational Biology.

As with all papers reviewed by the journal, your manuscript was reviewed by members of the editorial board and by several independent reviewers. The revision addressed many of the previous issues raised by the reviewers. However, the reviewers highlighted a number of concerns that still remain. Of particular importance, it is not clear whether the model proposed in the paper provides a better account of the data, or even can be distinguished from, alternative models - especially the linear threshold model proposed in Baumann et al., 2020. The revised paper includes a model comparison, however this appears to show little or no advantage of the subjective valuation model over the linearly declining threshold model.

Although the paper introduces a novel and interesting perspective on the stopping problem, in order to meet the standard for PLOS Computational Biology we feel it is important to demonstrate that the proposed model can explain aspects of behavior that are not already accounted for by existing models. Model comparison is one way to establish this. However an alternative approach, rather than comparing the quality of global fits, might be to identify and test specific features of behavior by which the models should be expected to differ.

In light of this issue, we would like to invite the resubmission of a significantly revised version, provided you can provide clear evidence in favor of the subjective optimality model over alternative models. The revised paper should also include a more complete discussion of alternative models, as well as taking into account the reviewers' other comments.

We cannot make any decision about publication until we have seen the revised manuscript and your response to the reviewers' comments. Your revised manuscript is also likely to be sent to reviewers for further evaluation.

Sincerely,

Adrian M Haith

Associate Editor

PLOS Computational Biology

Samuel Gershman

Deputy Editor

PLOS Computational Biology

Reviewer's Responses to Questions

**Comments to the Authors:**

Reviewer #1: The authors have done quite some work to address all the comments from the previous round of reviews. The paper is certainly improved. However, I am still not convinced that there is no selection bias here, nor am I convinced about the evidence in favor of the Prospect Theory (PT) model. These issues are linked. Because of the potential selection bias, it is unclear how much the thresholds really differ from “optimal”, and that is the crux of the argument for the PT model.

(1) The authors counter-argue that the threshold estimation was done at the individual level and therefore selection bias is not an issue. There is a problem with this argument, in that it assumes that all the variability in thresholds is at the subject level. But what if there is variability within-subject across trials? There is always some variability, the question is just how much. With any amount of variability, trials with low thresholds will disappear at earlier opportunities, leaving the higher threshold trials for later opportunities. It certainly seems like this could account for many of the patterns in the data. For instance, it is notable that there does not appear to be any deviation from optimality at the 1st opportunity. It is also notable that the deviation from optimality increases across opportunities. Both of these would I think be predicted by selection bias.

Another key set of results is on page 12, where the authors compare the threshold at 1/2, 4/5 and 9/10. There is clearly a difference between 1/2 and the others (which could easily be attributable to selection bias), but the difference between 4/5 and 9/10 is quite small, and seemingly non-significant (though the authors don’t report that comparison). So, overall the evidence here is not very compelling.

(2) Several of us asked for more evidence specifically in favor of the PT model. Here again the authors’ response was somewhat lacking. I do appreciate the model comparison, but the motivation and description of the new models was somewhat lacking; the entirety of it appears in a single supplemental figure. Moreover, the evidence isn’t that compelling. The advantage of the subjectively optimal model over the linear model is slight, and disappears once you remove the wait cost.

If we want to take this PT model seriously, shouldn’t it predict that people will be very unlikely to accept a number that is just below their threshold? Using the derived subjectively optimal thresholds for each subject, the authors could test whether there is a “gap” just below those thresholds where there are noticeably fewer acceptances than one would expect from symmetric noisy behavior around the thresholds. Ideally this would be done “out of sample” where the opportunity where this is tested is not used in fitting the subjectively optimal thresholds.

(3) I appreciate the RT analysis, but I think that for Figure 1c to be really compelling, the authors would want to compare to a version with the best-fitting constant threshold across opportunities. Otherwise it is difficult to judge whether these curves convincingly support dynamic over constant thresholds.

(4) It seems that model-based threshold predictions for K=5 are missing? I think this is because that was the dataset where the model was trained. However, that could be remedied by repeating the same exercise using a different dataset to train the model on, e.g. training on K=10 and then predicting on K=2 and K=5.

(5) Regarding the pupil analyses: I think these are improved, but there is one lingering question. Do these analyses include the last opportunity, i.e. 5 out of 5? That decision seems fundamentally different than the others, since subjects should accept any number. There is still a small trend in RT here, but the decisions are noticeably faster overall, and asymptote pretty quickly above value = 0. I wonder if Opportunity = 5 is overrepresented in the “Value-Threshold > 0”, which could explain why the pupil size appears so flat for acceptances compared to rejections in Figure 5B. Basically, it would be helpful to see this figure broken down by opportunity number, five panels in total. That would help to further support the claim that pupil dilation is different for acceptances compared to rejections.

Minor comments:

In what sense is Prospect Theory “heuristic”? It seems just as complex, if not more so, than expected utility. The authors might look at the recent work by Diederich & Trueblood (2018; Psychological Review) for one potential justification for this.

The authors might also be interested in Diehl & Poynor (2018), which shows that people have higher expectations when asked to choose from larger sets. They feel more disappointment for the same value item when it comes from a larger set than when it comes from a smaller set. This seems to parallel the current authors’ results.

Reviewer #2: I thank the authors for engaging with my previous comments and for the new analyses and revisions they have made to the manuscript. Unfortunately, my overall evaluation of the manuscript has not changed much. I continue to think that the main contribution is the novel empirical behavioral results, and to find the modeling less convincing.

1. Regarding my first comment on whether the assumptions of the dynamic programming modeling approach are justified, the new analyses suggest that there is not much learning happening, outside of experiment 1 where subjects do learn about the distribution of candidates, and/or that what learning does occur happens quickly. This is reassuring. However, the new analyses also show that there is an effect of previous candidate values (which were passed over) on current choices. The authors dismiss this finding as small, but the deviations from the optimal solution are also relatively small, and occur for exactly those decisions (3rd and 4th opportunities) where this is an effect of previous candidates. Therefore, it seems that a reasonable alternative (or addition) to subjective optimality model would that people are less likely to accept candidates that are above the optimal threshold but below a value that they’ve previously passed up. This kind of regret would result in subjects not adjusting their thresholds down as strongly as they should.

The model comparison also suggests that the subjective optimality model does not fit significantly better than a simple model that assumes people set a threshold and then linearly reduce that threshold as opportunities pass. It seems that such a heuristic model is another alternative to the subjective optimality model that cannot be ruled out by the data.

2. Regarding my second comment on the motivation for assuming prospect theory evaluation/a reference point, I take the authors’ point that empirically a fixed reference point in their model provides the best fit to the data. However, as I try to understand how the prospect theory pieces in the model account for behavior, I wonder whether the reference point is playing a major role or not. On page 10, the authors say that the subjective optimality model reduces to the optimal model if the value sensitivity parameter equals one (ie, if utility is a linear function of value). This suggests that the reference point is irrelevant if utility is linear. Could a model that assumes a non-linear utility function but has no reference point (expected utility) account for the data? What about a model that assumes a reference point plus loss aversion (linear utility with different slopes for gains and losses)? The broader point is about the motivation for the current model specification – the authors assume some pieces of prospect theory, but not others, and how those specific pieces are motivated by the data is not entirely clear.

3. Regarding my fourth comment on waiting cost, I take the authors’ point that changes in waiting cost across experiments fit the data. But the waiting cost is doing significant work in accounting for the major differences across experiments (whether thresholds are above the optimal in Experiments 1 and 2 or below in Experiment 3) and within experiments (in accounting for differences between K=5 and K=10 in Experiment 2). Yet waiting cost is not a part of the model that seems to be of theoretical importance in the paper. Why would waiting cost go up with more opportunities (K=10 in Experiment 2)? Is it really the delayed response requirement in Experiment 3 that increases waiting cost? Maybe the requirements of pupillometry (keeping head still, for example) make the subjects less tolerant of waiting? The current data can’t show for certain what is causing the differences in waiting cost. The paper is framed broadly in terms of the differences between an optimal model (expected value) and subjective optimality (prospect theory), but then the waiting cost parameter is doing much of the work.

4. Regarding my fifth comment on the pupil data, I thank the authors for clarifying the multiple comparisons correction for the association between pupil diameter and value sensitivity. It is still not clear to me why we would expect this relationship, though, and it is really the only link between the pupil data and the model/rest of the paper. The findings of pupil associations with choice and choice difficulty are (nice) replications of previous work, but they don’t hinge on the specifics of the subjective optimality model.

Reviewer #3: This is a very thorough and comprehensive revision, and I appreciate the authors' effort. All of my comments have been addressed.

**Have the authors made all data and (if applicable) computational code underlying the findings in their manuscript fully available?**

Reviewer #1: Yes

Reviewer #2: Yes

Reviewer #3: None

PLOS authors have the option to publish the peer review history of their article (what does this mean?). If published, this will include your full peer review and any attached files.

Reviewer #1: No

Reviewer #2: No

Reviewer #3: No
---

## [Decision Letter · Decision Letter 2]

12 Oct 2021

Dear Dr Chung,

Thank you very much for submitting your manuscript "Subjective optimality in finite sequential decision-making" for consideration at PLOS Computational Biology. As with all papers reviewed by the journal, your manuscript was reviewed by members of the editorial board and by several independent reviewers.

The reviewers have not reached a consensus on this paper. However, we feel that there is value in this work. The majority of the issues raised by the reviewers have been adequately addressed, and new analyses provide evidence that the prospect-theory-based model can account for aspects of behavior not captured by existing models.

However, as the reviewers noted, although the responses to the reviewers concerns were very thorough, many of these issues are still not addressed in the main text of the paper, despite being of critical importance in clearly establishing the contribution of the current paper. In particular, the comparison between the prospect-theory-based model and alternative models (e.g. the linear threshold model of Baumann et al.)  is critical evidence supporting the new model (and, indeed, is critical to our editorial decision on this paper). However, serious consideration of these alternative models is raised only in the Discussion as something of an afterthought, rather than as a key component of the argument in favor of the prospect-theory model.

Given the strengths of the work, we would be willing to accept the paper, pending some minor revisions. We would like you to ask you to revise parts of the main paper to more prominently address some the issues that arose during the review process. Specifically, we feel it is critical to critical to distinguish the prospect-theory-based model from alternative models as a part of the Results, rather than just including these in the Supplementary Information (Currently Figures S11-13). It would also further strengthen the paper to use the Discussion to clearly articulate other limitations raised during review (e.g. those pertaining to the wait cost, expected findings from pupillometry, etc.).

Sincerely,

Adrian M Haith

Associate Editor

PLOS Computational Biology

Samuel Gershman

Deputy Editor

PLOS Computational Biology

[LINK]

Reviewer's Responses to Questions

**Comments to the Authors:**

Reviewer #1: The authors have done an admirable job replying to my comments. I think it is clear that there is more going on here than just selection bias, regret, or linearly decreasing thresholds. I'm not totally convinced that the subjective optimality model is the singular best explanation, but it is grounded in solid theory (Prospect Theory) and seems to do better than any of the other proposed mechanisms on their own. In any case, the phenomenon is interesting and this paper will hopefully spur more work on this topic.

I do have one lingering concern that the authors should perhaps address in a minor revision. It has to do with the selection bias story. The authors treat the selection bias as a "model" and relegate it to the depths of the supplementary material (Figure S12) without any mention of it in the main text. I think this is a mistake. Selection bias is not a "model", it is a statistical phenomenon that is undoubtedly present in the authors' data. It is important to acknowledge and discuss it in the main text. Otherwise readers might overestimate the size of the deviation from "optimality" and future researchers might overlook this important detail of modeling such data. Personally, I would have liked to have seen a reasonable amount of selection bias incorporated into the other models (regret, linear threshold) and examined relative to the subjective optimality model in e.g. Fig. S11. However, I won't insist on this given where we are in the review process. But I would like to see more acknowledgment of this important statistical issue.

I also take issue with the results presented in Response Figure 2 (Fig. 12B) where the authors argue that the correlation between response variability and threshold change is considerably different in the data compared to the selection bias "model". This is obviously because of the assumptions in the authors' selection bias model. Specifically, the authors assume that that there is lots of across-round variability but no across-opportunities variability. In reality, there is surely both across-round and across-opportunity variability that are highly correlated. If that correlation was incorporated into the selection bias model, my guess is that it would more closely match the data. I would suggest that the authors either verify this, or simply omit Figure S12B from the paper. As it is now, I think that it is just misleading.

Reviewer #2: I appreciate the new analyses the authors have added in response to my comments, but I am afraid we’re at an impasse. While I feel that the issues raised over the last two rounds of review should have merited re-framing the paper, or tempering its conclusions substantially, the authors disagree. The main body of the paper has not really changed in this revision; additions have been made to the supplement but in many cases these are not even mentioned in the main text.

1. Regarding my previous comment that there was an effect of previous candidate values which were passed over on current choices, the authors present a model that adds to the optimal a feature that people do not accept candidates that are above threshold but below what they’ve previously passed up. This model (with no free parameters) also has the feature that thresholds do not decline as steeply as in the optimal model, but does not account for the data as well as the authors’ model (which has four free parameters). While this suggests that a regret is not the only feature necessary to account for the data, it does not show that regret is having no effect on people’s choices at all. The significant effects of past values on current choices is not accounted for by the subjective optimality model. But this weakness in the model is not acknowledged.

Similarly, the data the authors present to rule out the linear threshold reduction model is simply a re-plotting of the data shown in the main figure. The subjective optimality model traces the data better than the linear threshold reduction model (though mostly in the change between the third to last and second to last opportunity). But the subjective optimality model also has more parameters. When penalizing for this complexity, there is no difference between the two models.

2. Regarding my previous comment about the prospect theory specification (which has a reference point and non-linear utility), the authors present new analyses showing that other specifications (non-linear utility alone, or reference point plus linear utility plus loss aversion) do similarly. To my mind, what these fits reveal is that accounting for behavior requires adding to the optimal model a convex subjective transformation of the value space. The authors have reasons for favoring their transformation over others, but the specific convex transformation is not constrained by the data.

3. The authors did not make any changes in response to my previous comments about waiting cost. The waiting cost parameter still feels ad hoc to me (it’s not in Prospect Theory, for example, which is the inspiration for the model!), added to account for the differences between Experiment 3 and the others. But these results are discussed as if the differences in waiting cost were predicted by the model, or are a hypothesis confirmed by the model. To claim this as a prediction would require some hypothesis about what is driving the waiting cost that is directly tested.

4. The authors also did not make any changes in response to the previous comments about the pupil data. It is still not clear to me why we would expect higher pupil diameter in people with greater value sensitivity, or how this prediction arises from the subjective optimality model.

Reviewer #3: My original comments were addressed on the first revision, but the editors and other reviewers have made additional insightful questions, comments, and suggestions. Thus, my re-evaluation has focused on how the authors have addressed these issues.

Does the proposed model provide a better account of the data than existing models? In my view, the authors show that their model explains an aspect of behavior that is not accounted for by the linear threshold model. Specifically, they show that the slope of decision threshold decreases across opportunities. Although I don't fully understand why this relation is most pronounced in the last few opportunities, it seems like a clear advance over existing models.

I also think the authors were fairly thorough in their responses and revisions, but I defer to the other reviewers and editors about these specific issues since they were things that I did not personally raise. In any case, I appreciate the work that the authors and reviewers have put into this manuscript, and I think the product is much improved.

**Have the authors made all data and (if applicable) computational code underlying the findings in their manuscript fully available?**

Reviewer #1: Yes

Reviewer #2: Yes

Reviewer #3: Yes

PLOS authors have the option to publish the peer review history of their article (what does this mean?). If published, this will include your full peer review and any attached files.

Reviewer #1: No

Reviewer #2: No

Reviewer #3: No

Figure Files:

Data Requirements:

Reproducibility:

References:

---

## [Editor Report · Decision Letter 3]

11 Nov 2021

Dear Dr Chung,

We are pleased to inform you that your manuscript 'Subjective optimality in finite sequential decision-making' has been provisionally accepted for publication in PLOS Computational Biology.

Best regards,

Adrian M Haith

Associate Editor

PLOS Computational Biology

Samuel Gershman

Deputy Editor

PLOS Computational Biology

---

## [Editor Report · Acceptance letter]

24 Nov 2021

PCOMPBIOL-D-20-01425R3 

Subjective optimality in finite sequential decision-making

Dear Dr Chung,

I am pleased to inform you that your manuscript has been formally accepted for publication in PLOS Computational Biology. Your manuscript is now with our production department and you will be notified of the publication date in due course.

With kind regards,

Zsanett Szabo
